# Robust Pose Estimation in Crowded Scenes with Direct Pose-Level Inference

**Dongkai Wang**
Peking University
dongkai.wang@pku.edu.cn

**Shiliang Zhang**
Peking University
slzhang.jdl@pku.edu.cn

**Gang Hua**
Wormpex AI Research
ganghua@gmail.com

## Abstract

Multi-person pose estimation in crowded scenes is challenging because overlapping and occlusions make it difficult to detect person bounding boxes and infer pose cues from individual keypoints. To address those issues, this paper proposes a direct pose-level inference strategy that is free of bounding box detection and keypoint grouping. Instead of inferring individual keypoints, the Pose-level Inference Network (PINet) directly infers the complete pose cues for a person from his/her visible body parts. PINet first applies the Part-based Pose Generation (PPG) to infer multiple coarse poses for each person from his/her body parts. Those coarse poses are refined by the Pose Refinement module through incorporating pose priors, and finally are fused in the Pose Fusion module. PINet relies on discriminative body parts to differentiate overlapped persons, and applies visual body cues to infer the global pose cues. Experiments on several crowded scenes pose estimation benchmarks demonstrate the superiority of PINet. For instance, it achieves 59.8% AP on the OCHuman dataset, outperforming the recent works by a large margin[†].

## 1 Introduction

Multi-Person Pose Estimation (MPPE) performs both detection and pose keypoint localization for all persons appearing in an image. Due to its importance in human activity understanding, human-object interaction, human parsing, *et al.*, MPPE has attracted increasing attention in recent years. Current MPPE research can be summarized into two categories according to their pipelines, *i.e.* i) top-down methods [5, 18, 29, 25], which detect person bounding boxes and perform pose estimation for each bounding box, and ii) bottom-up methods [1, 20, 13, 17, 9, 7] that first detect body keypoints, then assign them into corresponding persons. A more detailed review will be presented in Sec. 2.

The above two lines of research have significantly boosted the performance of MPPE. However, performing MPPE is still challenging in crowded scenarios, where occlusions and person overlapping commonly occur. It is difficult to detect accurate bounding boxes due to overlapped persons. The commonly used Non-maximum Suppression (NMS) in detection frameworks also tends to suppress dense bounding boxes, leading to many undetected persons. Those issues degrade the accuracy of top-down methods. Person overlapping and occlusions lead to invisible keypoints, making it difficult for bottom-up methods to infer accurate pose cues from individual keypoints. As shown in many works, applying existing MPPE methods in crowded scenes gets inferior performance [10, 22].

As illustrated in Fig. 1 (a) and (b), existing works generally suffer from difficulties of person detection and keypoint grouping in crowded scenarios. This paper targets to perform MPPE with a new pipeline, which is free of bounding box detection and keypoint grouping. We regard the human pose as an inference objective, and directly infer the complete pose cues, *i.e.*, locations of all body keypoints, for a person from his/her visible body parts. In other words, we rely on discriminative visible body

---

[†]Code is available at: https://github.com/kennethwdk/PINet

35th Conference on Neural Information Processing Systems (NeurIPS 2021).

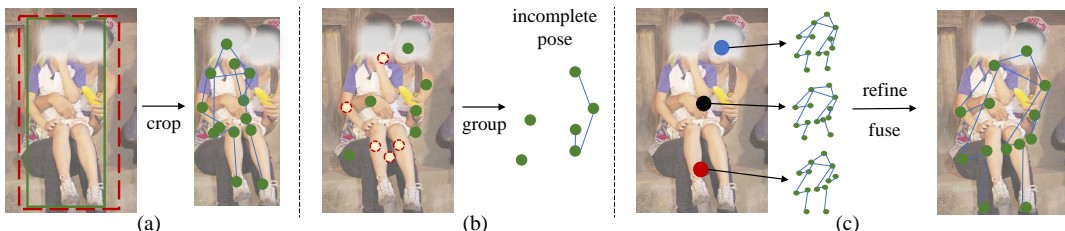

Figure 1: (a) Top-down methods suffer from the difficulty of detecting overlapped person in red bounding box; (b) Bottom-up methods are sensitive to keypoint occlusions in red circles, thus can not infer accurate pose cues; (c) The proposed DPLI infers initial pose cues from body parts, *e.g.*, three parts in the figure. Initial part cues are refined and fused to get the pose cues for the occluded person.

parts, *e.g.*, heads, to differentiate overlapped persons. We apply visible part cues and contexts to infer the global pose, including both visible and invisible keypoints. As this pipeline directly infers complete pose cues, we name it as the Direct Pose-Level Inference (DPLI). This idea is also inspired by works that infer invisible cues according to visible ones [24, 22, 30]. For instance, PRNet [24] detects occluded persons according to visible body parts and person bounding box ratio prior.

We implement DPLI with the Pose-level Inference Network (PINet) that progressively infers the occluded person pose from coarse to fine. Due to the lack of visible part annotations, we divide each person into several parts according to settings in Fig. 3 to acquire and exploit visible parts. The Part-based Pose Generation (PPG) generates initial pose cues from each part. The subsequent Pose Refinement model employs a dynamic graph convolution to leverage contexts and pose structure prior for the pose refinement. Finally, PINet uses Pose Fusion model to identify pose cues belonging to the same person, and fuse them as the final pose estimation.

We test our method on crowded scenes pose estimation benckmarks including OCHuman and CrowdPose. Experiments results show that, our PINet achieves superior performance compared with recent works. For instance, we achieve 59.8% AP on OCHuman, outperforming the recent OPEC-Net [22] and DEKR [4] by 30.7% and 7.8%. On CrowdPose, PINet achieves 62.2% AP on the hard test set mostly composed of crowded scenes. On this challenging test set, PINet outperforms the recent DEKR [4] by 4.7% in AP.

Some recent works also detect occluded keypoints from visible ones for pose estimation. OPEC-Net [22] detects occluded keypoints by applying a Graph Convolution Network (GCN) [32] on visible keypoints and contexts. However, OPEC-Net falls into the top-down category because it requires bounding boxes to discriminate persons, thus suffers from the difficulty of person detection. To the best of our knowledge, this is an original contribution on the Direct Pose-Level Inference (DPLI) for pose estimation in crowded scenarios, which differs from the previous top-down and bottom-up pipelines. The superior performance shown by extensive experiments validates the effectiveness of our proposed PINet and illustrates potentials of the DPLI pipeline.

## 2   Related Work

This work is related with multi-person pose estimation and crowded scenes pose estimation. This section briefly reviews related works in these categories respectively.

**Multi-Person Pose Estimation** aims to estimate the pose of every person in the image. Most of related works can be summarized into two categories, *i.e.*, top-down methods and bottom-up methods, respectively. Top-down methods first detect person bounding box by an object detector like YOLO [23], then perform single-person pose estimation in the cropped image or feature map. Mask R-CNN [5] is a typical method that directly adds a keypoint detection branch on Faster R-CNN to use the RoI pooling feature. G-RMI [18] and following methods break top-down methods into two steps and use separate models for person detection and pose estimation, respectively. SimpleBaseline [29] introduces a strong pose estimation baseline by adding deconvolution to ResNet [6]. Hourglass [14] and HRNet [25] propose new convolution network architectures to generate high quality and high resolution feature map for heatmap estimation, leading to superior performance.

Bottom-up methods first detect identity-free body keypoints for all persons in an image, then group those keypoints into individual persons. Heatmap is widely adopted for keypoint detection and most existing bottom-up methods focus on identifying and grouping keypoints belonging to the same person. DeepCut [20] is an early work that formulates grouping as an integer linear program. OpenPose [1] proposes the part affinity field approach to learn a vector field linking two keypoints. Grouping is conducted by calculating line integral between two keypoints and grouping the pair with the largest integral. Associative Embedding [13] learns each keypoint with a tag embedding and conducts the grouping by clustering the tag. PersonLab [17] groups keypoints by directly learning a 2D offset field for each pair of keypoints. PifPaf [9] extends OpenPose and PersonLab by using part intensity field to localize body parts, then applies part association field to associate body parts to infer the human pose. HGG [7] introduces a differentiable graph clustering to replace traditional grouping operation and obtains superior performance. Several recent works [16, 33, 15, 28] densely regress person poses from the person center point. Those methods suffer from issues of center point occlusion in crowded scenes, and difficulty of long distance and dense regression [16, 26].

**Crowded Scenes Pose Estimation** has not been fully exploited by the research community. There are few works to solve multi-person pose estimation when persons are highly overlapped. CrowdPose [10] constructs a large crowded pose estimation benchmark and proposes to apply global association to deal with crowded scenes. OPEC-Net [22] refines initial estimated pose by introducing the Graph Convolution Network. It requires bounding boxes to discriminate persons, thus suffers from the difficulty of person detection. The proposed PINet follows a novel Direct Pose-Level Inference pipeline, and is more robust to difficulties of person detection and keypoint grouping in crowded scenes.

## 3 Methodology

### 3.1 Overview

Given an image $I$ with multiple persons, the goal of crowded scenes pose estimation is to estimate locations of pose keypoints for each person, including both visible and invisible keypoints,

$$(\{\mathcal{K}^{(i)}\}, \{\bar{\mathcal{K}}^{(i)}\})_{i=1}^m = \mathrm{MPPE}(I)$$

where $\{\mathcal{K}^{(i)}\}$ and $\{\bar{\mathcal{K}}^{(i)}\}$ denote visible and invisible keypoints for $i$-th person, respectively. $m$ represents the number of persons in image $I$. We use $n = |\{\mathcal{K}^{(i)}\}| + |\{\bar{\mathcal{K}}^{(i)}\}|$ to denote the number of pose keypoints for each person, $e.g.$, $n = 17$ for OCHuman [31] and $n = 14$ for CrowdPose [10].

Unlike previous works that infer individual keypoint $\mathcal{K}$, PINet directly infers the complete pose cues $\{\{\mathcal{K}^{(i)}\}, \{\bar{\mathcal{K}}^{(i)}\}\}$ for each person. To achieve this goal, we adopt the Structure Pose Representation (SPR) [16] to represent the pose cue as a point $\mathbf{p}$ and a collection of offsets $\mathbf{o}$, $i.e.$,

$$\mathcal{P} = \{\mathbf{p}, \mathbf{o}\} = \{(x, y), (\delta x_k, \delta y_k)_{k=1}^n)\}, \tag{1}$$

where $(x, y)$ represents the coordinates of point $\mathbf{p}$ and $(\delta x_k, \delta y_k)$ denotes the offset of $k$-th pose keypoint to $\mathbf{p}$. According to Eq. (1), pose estimation can be conducted by detecting $\mathbf{p}$ and estimating offsets $\mathbf{o}$.

$\mathbf{p}$ should be carefully selected during the training stage, $e.g.$, locate on discriminate parts to infer high quality pose cues and avoid occlusions. SPR [16] uses the center point as $\mathbf{p}$, which could suffer from occlusions. We divide each person into $s$ parts and use each part center as $\mathbf{p}$. More detailed discussions on the body partition and the selection of $\mathbf{p}$ are presented in Sec. 3.2 and Fig. 3.

To locate $\{\mathbf{p}_j\}_{j=1}^s$ for each person and acquire corresponding $\{\mathbf{o}_j\}_{j=1}^s$, we propose the PPG to infer a confidence map and an offset map for each part, $i.e.$,

$$\{\mathbf{P}_j, \mathbf{O}_j\}_{j=1}^s = \mathrm{PPG}(\mathbf{F}), \mathbf{F} = \Phi(I), \tag{2}$$

where $\mathbf{P}_j$ represents the confidence map indicating the location of the $j$-th part, and $\mathbf{O}_j$ denotes the offset map, respectively, $\Phi(\cdot)$ is the backbone for feature extraction. Initial pose cues are acquired by applying the NMS on $\mathbf{P}_j$ to get multiple local maximum points $\mathbf{p}_j \in \mathbf{P}_j$ and sampling corresponding offsets $\mathbf{o}_j \in \mathbf{O}_j$. Each pair of $\{\mathbf{p}_j, \mathbf{o}_j\}$ composes an initial pose cue denoted by $\mathcal{P}'_j$.

The coarse pose $\mathcal{P}'_j$ can be refined by another PR module according to pose structure prior $G$ and contextual information included in the feature map $\mathbf{F}$, $i.e.$,

$$\hat{\mathcal{P}}_j = \mathrm{PR}(\mathcal{P}'_j, \mathbf{F}, G), \tag{3}$$

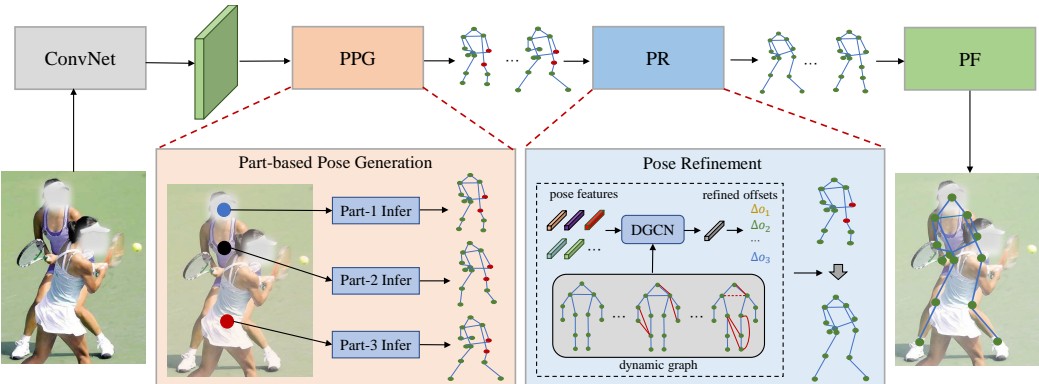

Figure 2: Framework of the proposed Pose-level Inference Network (PINet). We use a CNN to extract the feature map. Part-based Pose Generation (PPG) infers the whole person pose from different body parts, as denoted by circles in different colors. Pose Refinement (PR) refines the coarse person pose with dynamic pose keypoint graph. PF denotes the pose fusion operation that fuses the poses from different parts and output the final pose. We only show the pose of occluded person.

where $\hat{\mathcal{P}}_j$ denotes the pose refined by PR.

Because each part predicts a pose cue, there exist duplicate poses for the same person. We hence use a post-processing module PF to identify duplicate pose cues and fuse them as the final pose estimation,

$$\{\mathcal{P}^{(i)}\}_{i=1}^m = \text{PF}(\{\hat{\mathcal{P}}_j\}_{j=1}^t), \tag{4}$$

where $\mathcal{P}^{(i)}$ denotes the final pose estimation for the $i$-th person, $t$ is the number of poses generated by PPG and PR. PF can be implemented by measuring similarities between pose cues and merging similar poses.

We learn PINet by training the backbone $\Phi(\cdot)$, $\text{PPG}(\cdot)$, and $\text{PR}(\cdot)$ modules in an end-to-end manner. The overall training objective is denoted by

$$\mathcal{L} = \mathcal{L}_{PPG} + \lambda\mathcal{L}_{PR}, \tag{5}$$

where $\mathcal{L}_{PPG}$ and $\mathcal{L}_{PR}$ denote the loss for two modules and $\lambda$ is a weight to balance two losses. Fig. 2 illustrates the pipeline of proposed PINet. With the PPG, PR and PF, we can infer high quality person pose even when they are highly overlapping, and are free of missed keypoint and person detection in previous methods. In the following, we will present details of PPG, PR and PF.

## 3.2 Part-based Pose Generation

As stated above, we divide each person into $s$ body parts according to body skeleton to exploit the visible information. Generally speaking, the finer the body partition, the more likely there exists visible parts. However, small part is hard to regress full person pose offset due to the lack of global contexts. Moreover, too large $s$ also leads to the difficulty of dense regression optimization, which is discussed in [26].

To investigate a prior body partition strategy, we refer to previous work [11, 21] on pose keypoint and body parts and design several typical body partition strategies, as shown in Fig. 3. The partition strategies are listed from coarse to fine. For instance, *center* denotes that we divide person into one part, while *keypoint* indicates using 13 parts. To test the robustness of different partition strategies to occlusions, we show the occlusion ratio on OCHuman in Fig. 3. Note that, a partition without any visible parts is regard as an occlusion. As shown in the figure, finer body partition is more robust to occlusion. However, using too many parts is also harder for model optimization, which is verified in Sec. 4. We empirically find that part-large partition strategy achieves a good tradeoff. We hence divide each person into $s = 3$ parts, *i.e.*, upper, middle and bottom parts.

According to above analysis, PPG detects upper, middle and bottom parts for each person and regresses corresponding offsets. Following keypoint-based detection work [33], we detect each part

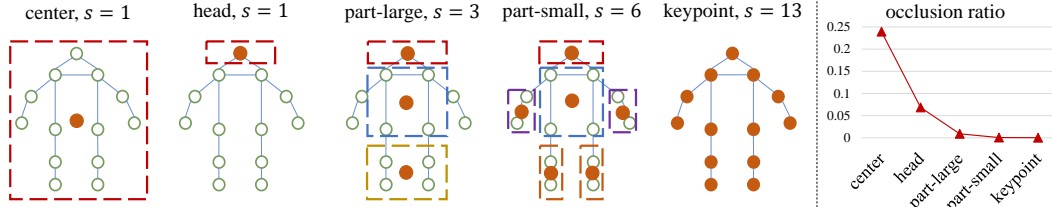

Figure 3: Illustration of different strategies of body partition in PPG: (a) Five different partition strategies, rectangles denote the body parts and brown circles denote part centers, and (b) their corresponding occlusion statistics from OCHuman dataset.

by estimating the part center with heatmap $\mathbf{P}$. Specifically, given the feature map $\mathbf{F} = \Phi(x)$ from backbone, we pass it through the keypoint detection head and offset regression head,

$$\mathbf{P}_j = \mathcal{F}_j(\mathbf{F}), \mathbf{O}_j = \mathcal{R}_j(\mathbf{F}), j \in \{u, m, b\} \tag{6}$$

where $\mathbf{P}_j, \mathbf{O}_j$ represent the confidence map of $j$-th part and corresponding offset map, respectively. $u, m, b$ denote three parts. We adopt the tradeoff $l_2$ regression loss to learn $\mathbf{P}_j$ and use smooth $l_1$ loss to learn the offset map $\mathbf{O}_j$,

$$\mathcal{L}_{point} = ||\mathbf{M} \odot (\mathbf{P} - \mathbf{P}^*)||_2^2, \tag{7}$$

$$\mathcal{L}_{offset} = \frac{1}{|\Omega|} \sum_{i \in \Omega} \frac{1}{S_i} \text{smooth}_{l_1}(\mathbf{o}_i - \mathbf{o}_i^*), \tag{8}$$

where superscript $^*$ indicates the groundtruth label. $\mathbf{P} = [\mathbf{P}_u; \mathbf{P}_m; \mathbf{P}_b]$ denotes the point maps, $\mathbf{M}$ is a mask to down weight background region, the weight of positions not lying in part center region is 0.1, and others are 1. $\Omega$ denotes the index set of part center and its neighbor points on feature map, $S_i$ is the area of $i$-th person. We generate the ground truth following [16]. The loss of PPG is the weight sum of above two losses, i.e., $\mathcal{L}_{PPG} = \mathcal{L}_{point} + \lambda_o \mathcal{L}_{offset}$. $\lambda_o$ is a weight to balance to losses.

### 3.3 Pose Refinement

Although PPG can generate complete person pose, it still confronts several issues. Firstly, the part-based generation strategy enhance the robustness of PINet to occlusion, but eliminate the person global information for each part to infer the whole person pose. i.e., the upper body part feature is hard to encode foot due to limited receptive field. Secondly, the occluded pose keypoint should be inferred by considering keypoint connections, namely pose structure, not only visual information.

Thus we are motivated to propose a PR module to address above issues. For pose $\mathcal{P}_j'$ generated by point feature $f_p$ in PPG, we supply it with keypoint features $f_k$ to encode the person global information, which can be obtained by bilinear sampling on the feature map $\mathbf{F}$ with $k$-th keypoint location of pose $\mathcal{P}_j'$. To fully utilize pose structure pose prior, we propose a dynamic graph convolution network (DGCN) that considers keypoint connection information to refine and aggregate these keypoint features,

$$X^{l+1} = \sigma(GX^lW), \tag{9}$$

where $W$ is the parameters of DGCN, $\sigma$ denotes ReLU, $X^l$ and $X^{l+1}$ is the original and updated features from $l$-th layer, and $G$ is the dynamic graph, which is initialized according to the human body skeleton and updated as the model parameters. We adopt 3 layers DGCN with residual connection to refine these features. Each row of the initial feature matrix $X^0$ represents one keypoint. Following [27], we apply max pooling and average pooling to the refined keypoint features to obtain the global features of the pose, which is concatenated with $f_p$ and passed to two fully connected layers to predict the offset $\Delta \mathcal{P}_j'$ of initial pose $\mathcal{P}_j'$ and the ground truth. Thus we can obtain the refined pose by $\hat{\mathcal{P}}_j = \mathcal{P}_j' + \Delta \mathcal{P}_j'$. Similar to the offsetmap loss, we calculate the refinement loss by,

$$\mathcal{L}_{PR} = \frac{1}{|\mathcal{T}|} \sum_{\mathcal{P}_j' \in \mathcal{T}} \frac{1}{S_i} \text{smooth}_{l_1}(\mathcal{P}_j' + \Delta \mathcal{P}_j', \mathcal{P}_j^*), \tag{10}$$

where $\mathcal{T}$ is the sampled pose set for PR refinement, $S_i$ is the area of $i$-th person.

**Discussions.** Previous OPEC-Net [22] also utilize GCN. However, PR is different from it in several aspects. Firstly, OPEC-Net focuses on inferring occluded keypoints and takes keypoint coordinates as input, thus require extra feature adaptation module to extract context information. Our PR focuses on extracting better person global information with pose structure prior and does not require extra module. Secondly, PR adopts a dynamic graph to refine keypoint features and can capture more keypoint relationship. For example, the skeleton contains no connections between left and right body. However, when the right ankle is occluded, we could refer to left ankle and knee for inference. The dynamic graph is more flexible and can capture these properties with model learning. Fig. 2 illustrates an example of the dynamic graph during training.

### 3.4 Pose Fusion

PPG generates duplicate poses and low-quality poses from invisible body parts, thus a pose-processing operation is required to eliminate them. However, duplicate poses from different body parts perform well on different keypoints due to long distance regression [16, 33], thus we can fuse them to get better results. A pose regressed from upper body parts performs well on locating keypoints on nose, while the pose inferred from the bottom body part gets poor performance on locating those keypoints.

According to above analysis, we propose a PF operation to fuse duplicate poses and remove low-quality poses. Given the poses inferred by PPG and PR, $i.e.$, $\{\hat{\mathcal{P}}_j\}_{j=1}^{t}$, PF firsts find the pose set that belong to the same person. Inspired by OKS-NMS [18], we adopt a OKS based matching strategy to identify the poses belonging to the same person. For two poses $\hat{\mathcal{P}}_1$ and $\hat{\mathcal{P}}_2$, we calculate their object keypoint similarity by,

$$\text{OKS}(\hat{\mathcal{P}}_1, \hat{\mathcal{P}}_2) = \frac{1}{n} \sum_{k=1}^{n} \exp\left(-\frac{||\hat{\mathcal{P}}_1[k] - \hat{\mathcal{P}}_2[k]||_2^2}{(S_1 + S_2)\alpha_k^2}\right) \tag{11}$$

where $\hat{\mathcal{P}}[k]$ denotes the $k$-th keypoint, $S_1$ and $S_2$ represent the area of two poses, $\alpha_k$ is a per-keypoint constant that controls falloff. If the OKS of two poses is larger than $\gamma$, they are regraded as duplicate poses. According to above matching strategy and NMS procedure, we can divide $\{\hat{\mathcal{P}}_j\}_{j=1}^{t}$ into $m$ groups, where each group $\{\hat{\mathcal{P}}_j\}_{j=1}^{t_i}$ represent $i$-th person pose set. Then PF traverse each pose set $\{\hat{\mathcal{P}}_j\}_{j=1}^{t_i}$ and select $n$ keypoints with maximum score to construct the final person pose $\mathcal{P}^{(i)}$. The detailed process can be seen in the supplementary material.

The $k$-th keypoint score is formed by $r = r_p \cdot r_k$, where $r_p$ is the confidence score of point from $\mathbf{P}$, $r_k$ is the keypoint confidence score which can be obtained by learning an extra score map $\mathbf{H}$ for each pose keypoints, $i.e.$, the heatmap in most MPPE methods. A low-quality pose generated by PPG has small $r_p$, thus will not appear in the final results. Moreover, $r_k$ is helpful in selecting well estimated keypoints and reliefing the long regression problem. During implementation, we learn the score map with the point map $\mathbf{P}$ and change Eq. 6 to,

$$\mathcal{L}_{point} = ||\mathbf{M} \odot ([\mathbf{P}; \mathbf{H}] - [\mathbf{P}^*; \mathbf{H}^*])||_2^2, \tag{12}$$

where $[\cdot; \cdot]$ denotes concatenation operation.

**Discussions.** PF is a model-agnostic operation and can also be applied to other regression methods, $i.e.$, [16]. However, it is more effective in our method when duplicate predictions are from different parts, not merely center points. We also validate the effectiveness of PF on other methods in Sec.4.

## 4 Experiments

### 4.1 Datasets and Evaluation Protocol

**OCHuman** [31] is a recently proposed benchmark to examine the limitations of MPPE in highly challenging scenarios. It consists of 4731 images in total, including 2500 images for validation and 2231 images for testing. The average IoU of the bounding boxes is 67%. Following [22], we train models on `val` set and report the performance on `test` set.

**CrowdPose** [10] is another dataset that includes many crowded scenes. It contains 10K, 2K and 8K images for `train`, `val` and `test` set. Following [2, 4], we train our models on the `train` and `val`

Table 1: Ablation studies of our proposed PINet on individual components.

(a) Effectiveness of each component in PINet.

| PPG | PR | PF | AP | AP$^{50}$ | AP$^{75}$ |
|---|---|---|---|---|---|
| | | | 47.6 | 67.5 | 53.2 |
| ✓ | | | 53.1 | 70.5 | 60.4 |
| ✓ | ✓ | | 58.8 | 74.3 | 65.5 |
| ✓ | | ✓ | 55.8 | 72.7 | 62.3 |
| ✓ | ✓ | ✓ | **59.8** | **74.9** | **65.9** |

(b) Test different body partition strategies in PPG.

| Part partition | AP | AP$^{50}$ | AP$^{75}$ |
|---|---|---|---|
| Center | 47.6 | 67.5 | 53.2 |
| Head | 49.8 | **70.9** | 55.6 |
| Part-large | **53.1** | 70.5 | **60.4** |
| Part-small | 50.3 | 69.4 | 57.2 |
| Keypoint | 40.8 | 59.0 | 45.7 |

(c) Test different refine strategies in PR.

| Method | AP | AP$^{50}$ | AP$^{75}$ |
|---|---|---|---|
| No GCN | 56.8 | 74.1 | 63.0 |
| Static graph | 57.7 | 74.7 | 64.3 |
| Dynamic graph | **59.8** | **74.9** | **65.9** |

(d) Test PF in different methods.

| Method | w/o PF | w/ PF |
|---|---|---|
| SPM [16] | 47.6 | 48.1 |
| DEKR [4] | 52.2 | 53.0 |
| PINet(w/o PR) | 53.1 | 55.8 |

sets and report the results on the `test` set. Since CrowdPose contains both crowded and non-crowded scenes, we also report the performance on crowded test set, namely metric $AP^H$.

**COCO** [12] is the popular MPPE benchmark that contains over 200,000 images. COCO keypoint benchmakr contains `train`, `val` and `test-dev` set. The `train` set includes 57K images and 150K person instances annotated with 17 keypoints, the `val` set contains 5K images, and the `test-dev` set consists of 20K images. However, it includes few crowded scenes, and we use it to valid the generation ability of our method to non-occluded scenes. We train the models on the `train` set and report the results on the `val` and `test-dev` sets. Moreover, following [7], We also directly test the model trained on COCO in OCHuman `val` and `test` set.

**Evaluation Metrics.** We follow the standard evaluation metric and use OKS-based metrics for pose estimation. We report average precision scores with different thresholds: AP, $AP^{50}$, and $AP^{75}$. On CrowdPose, we also report $AP^E$, $AP^M$ and $AP^H$, which refers to the performance under different crowd index.

## 4.2 Implementation Details

All experiments are implemented on PyTorch [19]. We adopt ImageNet [3] pre-trained HRNet-W32 [25] as backbone for all experiments and follow the most configuration of [2, 4]. We set $\lambda$ and $\lambda_o$ to 0.03 and $\gamma$ to 0.7 for all experiments. For PR, we sample 40 poses from PPG for each part.

**Training**. The input image is resized to 512*512. We use Adam [8] to optimize the model, the learning rate is set to 0.001 for all layers. We train the model for 140 epochs on OCHuman and COCO, with learning rate dividing by 10 at 90th, 120th epoch. For CrowdPose, we train model with 300 epochs and divide learning rate by 10 at 200th, 260th epoch. The batch size is set to 20 OCHuman and 40 for CrowdPose and COCO. We adopt data augmentation strategies including random rotation (-30,30), scale ([0.75,1.5]), translation ([-40,40]) and flipping (0.5).

**Testing**. We resize the short side of the images to 512 and keep the aspect ratio between height and width. We adopt single and multi-scale ([0.5,1.0,2.0]) test with flipping following [2, 4].

**Comparison methods**. To make fair comparison, we report the performance of SPM [33](with SPR), HigherHRNet [2] and DEKR [4] in the same setting, *i.e.*, backbone, training schedule and test setting.

## 4.3 Ablation Study

**Components analysis.** We analyze the effectiveness of each proposed components on OCHuman. The results are shown in Table 1 (a). PINet with only PPG achieves 53.1% AP, which is already higher than previous center-based regression work. This indicates that inferring complete pose from visible part is important for occluded pose estimation. PR incorporate pose structure prior to refine the coarse, it remarkably enhances the AP by 5.7%. We also evaluate the propose pose fusion operation, and it

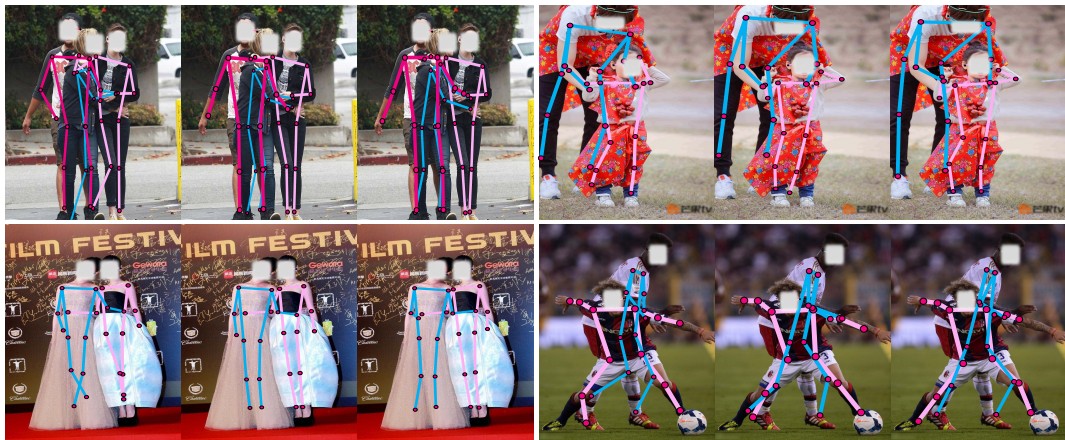

Figure 4: Qualitative results on OCHuman `test` set. Each image shows the prediction results of PPG, PPG+PR and PPG+PR+PF. It can be observed that both PR and PF can improve the accuracy of keypoint localization.

Table 2: Comparison with state-of-the-art methods on OCHuman `test` set. ∗ denotes our implementation based on the author's public code.

| Method | Reference | AP | $AP^{50}$ | $AP^{75}$ |
|---|---|---|---|---|
| Mask RCNN [5] | ICCV2017 | 20.2 | 33.2 | 24.5 |
| SimpleBaseline [29] | ECCV2018 | 24.1 | 37.4 | 26.8 |
| SPPE+ [10] | CVPR2019 | 27.6 | 40.8 | 29.9 |
| OPEC-Net [22] | ECCV2020 | 29.1 | 41.3 | 31.4 |
| SPM∗ [16] | ICCV2019 | 47.6 | 67.5 | 53.2 |
| HigherHRNet∗ [2] | CVPR2020 | 27.7 | 66.9 | 15.9 |
| DEKR∗ [4] | CVPR2021 | 52.2 | 69.9 | 56.6 |
| Ours | this paper | **59.8** | **74.9** | **65.9** |

boosts the performance by 1.0%, which validates the effectiveness of fusing duplicate predictions. As shown in Fig. 4, both PR and PF can improve the accuracy of keypoint localization.

**Analysis on PPG.** An important factor of PPG is the choice of body partition strategy. We compare different partition strategies in Fig. 3. Table 1 (b) summarizes the results. We can observe that center part achieves 47.6% AP, while head part obtains 49.8% AP. This is because that center part is usually occluded by other person, thus PPG cannot generate high quality pose. While head and face are discriminative regions, thus is more likely to be visible. The proposed part-large partition achieves the best performance, obtaining 53.1% AP, outperforming center and head points by 5.5% and 3.3% AP, respectively. We also investigate finer point selection, *i.e.*, part-small and keypoint.However, they work badly due to the hard optimization with dense regression.

**Analysis on PR.** We also analyze the component of PR. As shown in Table 1 (c). We first test to directly refine estimated pose by PPG without incorporating pose structure prior, and it obtains 56.8% AP. Incorporating pose keypoint graph to refine feature achieves 57.7% AP. It indicates that pose structure prior is useful for occluded pose inference. Moreover, when adding dynamic graph into PR, we can further boost AP to 59.8% AP, which verify the effectiveness of proposed methods.

**Analysis on PF.** To verify that PF is a model-agnostic operation, we apply it to other regression-based methods, *i.e.*, SPM and DEKR. As shown in Table 1 (d), PF can constantly boost these methods about 0.6-0.8 AP, which demonstrates the effectiveness of PF on regression methods. Moreover, PF is more useful when we regress pose from different parts, and it boost the performance of PINet from 53.1 to 55.8. Note that PR can also relieve the regression problem, thus we omit it here to verify PF.

Table 3: GFLOPs and #parameters of the representative top competitors and our approaches. ∗ denotes our implementation.

|  | SPM∗ [16] | HigherHRNet [2] | DEKR [4] | PINet |
|---|---|---|---|---|
| Input size | 512 | 512 | 512 | 512 |
| #param.(M) | 29.3 | 28.6 | 29.6 | 29.4 |
| GFLOPs | 41.8 | 47.7 | 45.4 | 42.4 |

Table 4: Comparison with state-of-the-art methods on CrowdPose `test` set. ∗ denotes our implementation. † denotes multi-scale test.

| Method | Reference | AP | $AP^{50}$ | $AP^{75}$ | $AP^E$ | $AP^M$ | $AP^H$ |
|---|---|---|---|---|---|---|---|
| Mask RCNN [5] | ICCV2017 | 57.2 | 83.5 | 60.3 | 69.4 | 57.9 | 45.8 |
| SimpleBaseline [29] | ECCV2018 | 60.8 | 81.4 | 65.7 | 71.4 | 61.2 | 51.2 |
| SPPE+ [10] | CVPR2019 | 66.0 | 84.2 | 71.5 | 75.5 | 66.3 | 57.4 |
| OpenPose [1] | CVPR2017 | - | - | - | 62.7 | 48.7 | 32.3 |
| SPM∗ [16] | ICCV2019 | 63.7 | 85.9 | 68.7 | 70.3 | 64.5 | 55.7 |
| HigherHRNet [2] | CVPR2020 | 65.9 | 86.4 | 70.6 | 73.3 | 66.5 | 57.9 |
| HigherHRNet† [2] | CVPR2020 | 67.6 | 87.4 | 72.6 | 75.8 | 68.1 | 58.9 |
| DEKR [4] | CVPR2021 | 65.7 | 85.7 | 70.4 | 73.0 | 66.4 | 57.5 |
| DEKR† [4] | CVPR2021 | 67.0 | 85.4 | 72.4 | 75.5 | 68.0 | 56.9 |
| Ours | this paper | 68.9 | 88.7 | 74.7 | 75.4 | 69.6 | 61.5 |
| Ours† | this paper | **69.8** | **89.1** | **75.6** | **76.4** | **70.5** | **62.2** |

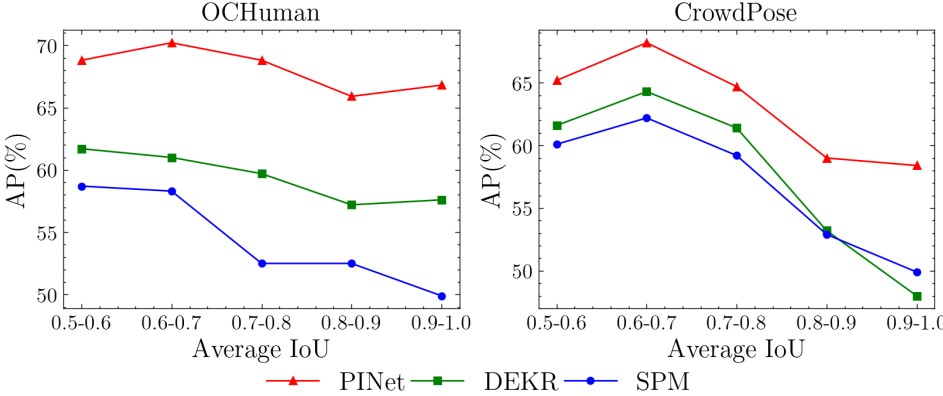

Figure 5: Comparison with DEKR, SPM and our proposed PINet under different IoU threshold. Our method constantly outperforms other methods, especially at highly crowded scenes (high IoU).

## 4.4 Comparison with Recent Works

We compare PINet with recent works on OCHuman [31], CrowdPose [10] and COCO [12]. The results are shown in Table 2, 4 and 5. Table 2 reports comparisons on OCHuman. We compare top-down methods: Mask RCNN [5], SimpleBaseline [29], SPPE+ [10] and OPEC-Net [22], and bottom-up methods SPM [16], HigherHRNet [2] and recent DEKR [4]. On OCHuman, top-down methods achieve relatively low AP. This is due to the issues mentioned in Sec 1. Our PINet adopts a pose-level inference, thus achieves the best 59.8% AP on OCHuman in single-scale test, which is higher than all bottom-up methods under the same setting. For instance, DEKR obtains 52.2% AP, which is lower than PINet by 7.6%.

On CrowdPose dataset, our method also obtains superior performance. As shown in Table 4, PINet achieves 68.9% and 69.8% AP in single-scale and multi-scale test, which is higher than HigherHRNet and DEKR. Moreover, on crowded scenes, PINet achieves 61.5% and 62.2% $AP^H$, which is higher

Table 5: Comparison with state-of-the-art methods on COCO and OCHuman. Models are trained on COCO `train` set.

| Method | Reference | COCO | | OCHuman | |
|---|---|---|---|---|---|
| | | val | test-dev | val | test |
| OpenPose [1] | CVPR2017 | - | 61.8 | - | - |
| AE [13] | NeurIPS2017 | - | 56.6 | 32.1 | 29.5 |
| PersonLab [17] | ECCV2018 | 66.5 | 66.5 | - | - |
| HGG [7] | ECCV2020 | 60.4 | - | 35.6 | 34.8 |
| HigherHRNet [2] | CVPR2020 | 67.1 | 66.4 | - | - |
| DEKR [4] | CVPR2021 | **68.0** | **67.3** | 37.8 | 36.4 |
| ours | this paper | 67.4 | 66.7 | **38.3** | **37.2** |

than DEKR by 4.0% and 5.3%, respectively. Results on two occluded datasets demonstrate that PINet is effective to deal with crowded scenes pose estimation.

To further investigate the robustness of our method to crowded scenes, we follow previous work [22] to split datasets according to the average IoU of bounding box. Since there are few or no images with IoU smaller than 0.5 on OCHuman, we start from 0.5 and divide the `test` set of OCHuman and CrowdPose into 5 parts and report the performance at each IoU. As shown in Fig. 5, our method constantly outperforms DEKR and SPM, especially at high IoU. We also report the parameters and GFLOPs of existing bottom-up methods, as shown in Table 3. Our method requires less or similar parameters than other methods but shows better performance.

We also report the performance on COCO dataset, the results are shown in Table 5. We train PINet on COCO `train` set, and directly test it on COCO and OCHuman. PINet achieves 67.4% AP on COCO `val` set, which is comparable to recent methods, *i.e.*, HigherHRNet [2] with 67.1%AP and PersonLab [17] with 66.5% AP, which indicates that PINet also works well on non-occluded scenes. Moreover, following [7], We also directly test the model trained on COCO in OCHuman `val` set and `test` set, and obtained 38.3% AP, which is higher than HGG by 2.7% and DEKR by 0.5%. This result demonstrates the generalization ability of proposed methods and its effectiveness to handle crowded scenes.

## 5 Discussion and Conclusion

In this paper we address the multi-person pose estimation in crowded scenes. Our proposed PINet has shown considerable improvements in crowded scenes pose estimation. This is achieved by the direct pose-level inference pipeline that infers pose by leveraging both body part cues and pose structure prior. Experiments on several crowded scenes pose estimation benchmarks demonstrate the effectiveness of proposed methods to handle occlusion. However, the performance of crowded scenes is still lower than non-crowded scenes, and PINet directly regresses the pose, which is not as accurate as the heatmap-based methods. Further studies are required to address these problems.

## Acknowledgements

This work is supported in part by Peng Cheng Laboratory, in part by The National Key Research and Development Program of China under Grant No. 2018YFE0118400, in part by Natural Science Foundation of China under Grant No. U20B2052, 61936011, 61620106009, in part by Beijing Natural Science Foundation under Grant No. JQ18012.

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
