# Robust Pose Estimation in Crowded Scenes with Direct Pose-Level Inference
## Supplementary Materials

**Dongkai Wang**
Peking University
dongkai.wang@pku.edu.cn

**Shiliang Zhang**
Peking University
slzhang.jdl@pku.edu.cn

**Gang Hua**
Wormpex AI Research
ganghua@gmail.com

## 1 Groundtruth Generation

In Sec. 3 we present the overall objective of PINet, here we provide more details about the groundtruth used for PINet training[†]. Following previous work [2], we generate gaussian heatmap for point map $\mathbf{P}^*$ and score map $\mathbf{H}^*$ learning. Given the part center point or pose keypoint $p = (x, y)$, the corresponding heatmap can be generated by,

$$\mathbf{P}^*(i, j) = \exp(-||(i, j) - (x, y)||_2^2 / \sigma^2), \tag{1}$$

where $\sigma$ is a constant to control the variance of gaussian distribution, we set it to 2 for all points. $\mathbf{H}^*$ can be generated in the same way. If there exists multiple persons in an image, then there are multiple peaks in a heatmap. Here we also follow [2] to take the maximum of confidence maps.

As for offset map $\mathbf{O}^*$, we construct a dense displacement map. For $k$-th part with center point $\mathbf{p}_k = (x, y)$, we calculate the offset map by,

$$\mathbf{O}^*(i, j) = \begin{cases} \mathbf{o}_k & \text{if } (i, j) \in \mathcal{N}_k \\ 0 & \text{otherwise} \end{cases} \tag{2}$$

where $\mathcal{N}_k = \{(i, j) | ||(i, j) - (x, y)||_2^2 \le r\}$ denotes the neighboring positions of the part center point, $\mathbf{o}_k$ is the offset of pose according to $\mathbf{p}_k$. We construct three offset maps for upper, middle and bottom part according to Sec. 3.2 in original paper. $r$ is set to 2 for all points. And the offset loss are only calculated with the non-zero points in offset map, which is denoted by $\Omega$ in Eq.(10). We also will generate a weight map $\mathbf{S}$ that represent the area of persons,

$$\mathbf{S}(i, j) = \begin{cases} S_k & \text{if } (i, j) \in \mathcal{N}_k \\ 0 & \text{otherwise} \end{cases} \tag{3}$$

where $S_k$ denotes the area of $k$-th person.

## 2 Pose Sampling Strategy for PR

Here we give the details of how to sample pose for PR refinement. In training we sample pose according to the ground truth. Given the point heatmap $\mathbf{P}^*$ and corresponding weight map $\mathbf{S}$, we only sample feature points $(i, j)$ that has non-zero value on weight map. If two feature points are satisfied, then we select the points with larger value on point heatmap $\mathbf{P}^*$. We sample 40 points for each part in training, and thus 120 poses in total are sent to PR.

During inference, we first follow [1] and apply a max pooling to find the local maximum points on predicted point heatmap, then sample corresponding offset map. The number of sampled points is set to 30 for each part point map.

---

[†]https://github.com/kennethwdk/PINet

35th Conference on Neural Information Processing Systems (NeurIPS 2021).

# 3 Algorithm for Pose Fusion

In this section we provide a detailed algorithm description for Pose Fusion (PF) operation, as shown in Alg. 1.

---

**Algorithm 1** Pose Fusion

---

**Input**: The estimated pose set $\{\hat{\mathcal{P}}_j\}_{j=1}^t$. Confidence score $\{r_p^j\}_{j=1}^t$ for each estimated pose. Pose keypoint score map $\mathbf{H}$. Pose keypoint number $n$.
**Output**: Fused final pose set $\{\mathcal{P}^{(i)}\}_{i=1}^m$, $m$ represents the number of persons.

1: Initialize fused pose set $\{\mathcal{P}^{(i)}\} = \{\}$;
2: Sort $\{\hat{\mathcal{P}}_j\}_{j=1}^t$ according to confidence score $\{r_p^j\}_{j=1}^t$ from high to low to get $\{(\hat{\mathcal{P}}_j, r_p^j)\}_{j=1}^t$;
3: **for** $j$ **in** $1...t$ **do**
4:     Get $j$-th pose $\hat{\mathcal{P}}_j$ and confidence score $r_p^j$;
5:     **if** $\hat{\mathcal{P}}_j$ is used **then**
6:         continue;
7:     **end if**
8:     Match $\hat{\mathcal{P}}_j$ to the not used pose set according to Eq. 11 and get the matched pose set $\{\hat{\mathcal{P}}_j\}_{j=1}^{t_i}$;
9:     Mark $\{\hat{\mathcal{P}}_j\}_{j=1}^{t_i}$ as used;
10:    Initialize fused pose $\mathcal{P}^{(i)} = \hat{\mathcal{P}}_j = \{\mathcal{K}_j^k\}_{k=1}^n$;
11:    **for** $k = 1 \cdots n$ **do**
12:       Compute the keypoint score $r = r_p \cdot \mathbf{H}_k[\hat{\mathcal{P}}[k]]$ for each pose in $\{\hat{\mathcal{P}}_j\}_{j=1}^{t_i}$;
13:       Select the keypoint with maximum $r$ and set it to $\mathcal{P}^{(i)}[k] = \mathcal{K}_{max}^k$;
14:    **end for**
15:    Add $\mathcal{P}^{(i)}$ to the fused pose set $\{\mathcal{P}^{(i)}\}$;
16: **end for**
17: **return** $\{\mathcal{P}^{(i)}\}_{i=1}^m$

---

# 4 More Visualization Results

We visualize some crowded scenes pose estimation results on both OCHuman and CrowdPose, as shown in Fig. 1 and Fig. 2. Even two persons are highly overlapping and occluded, our PINet can still output the corresponding person poses.

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

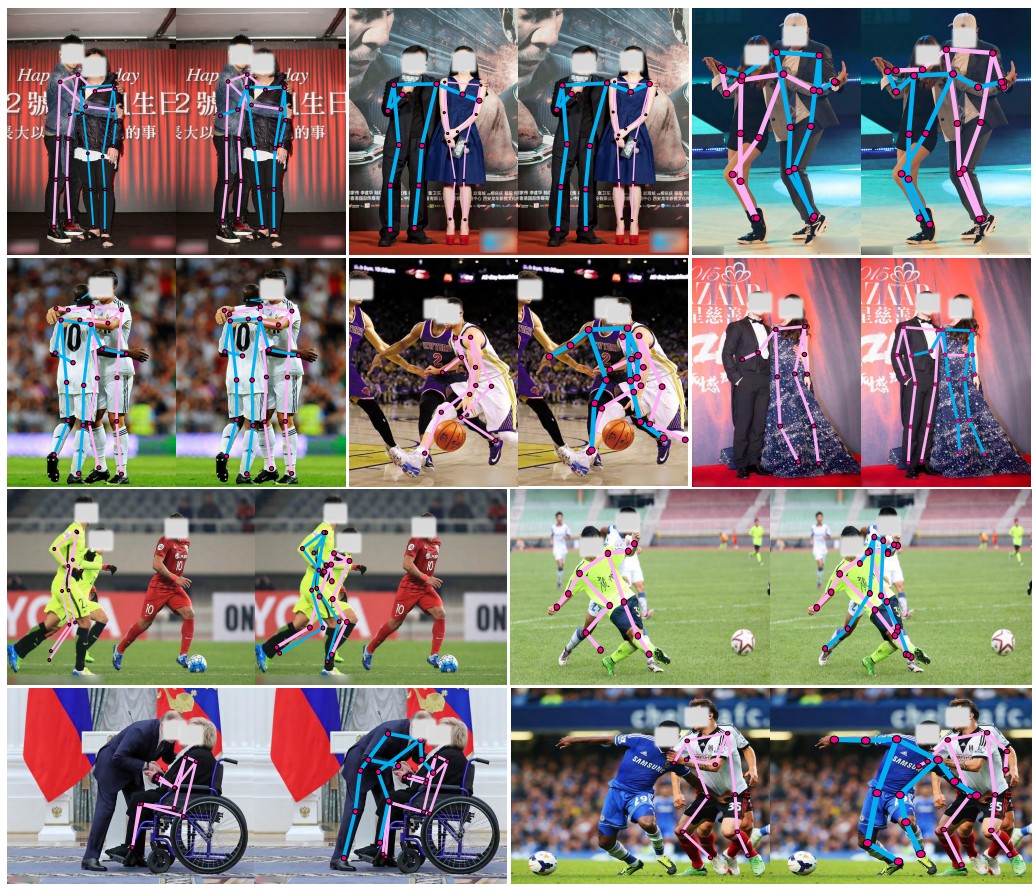

Figure 1: Qualitative results on OCHuman. Each image (left to right) shows DEKR predictions and our PINet predictions. Due to occlusions, DEKR often misses the occluded persons which are recovered by PINet.

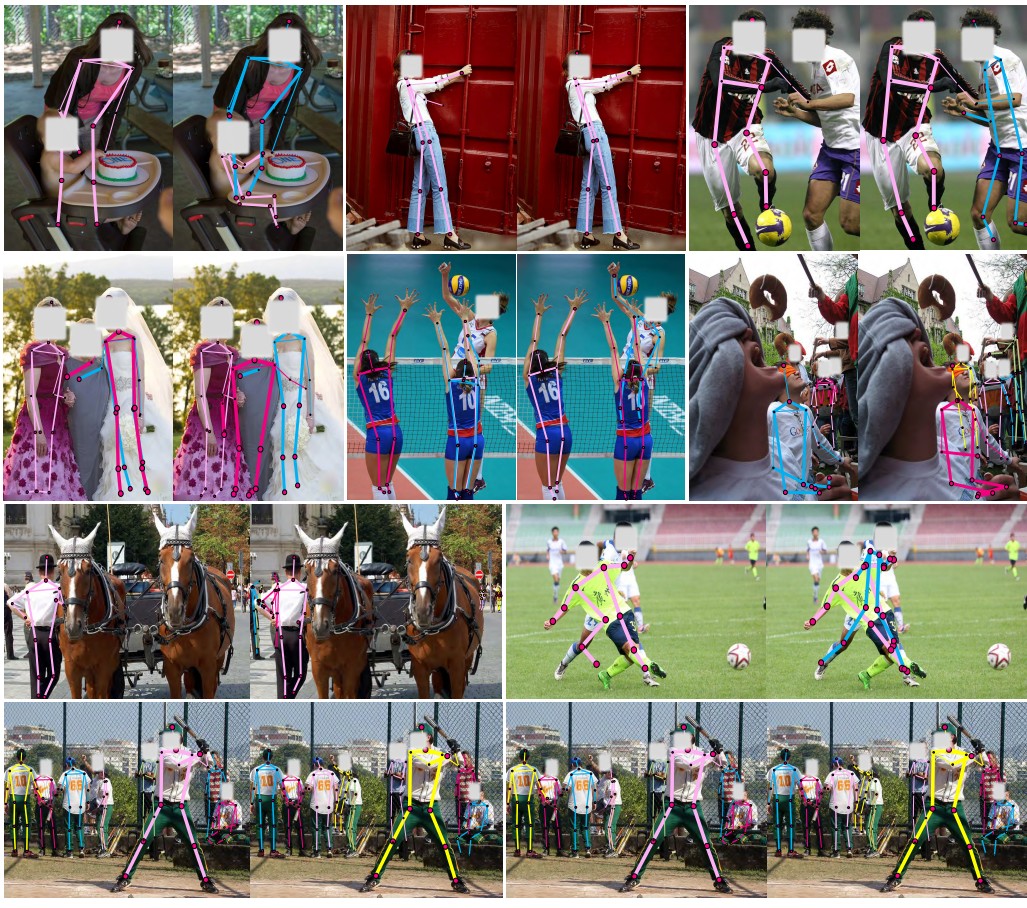

Figure 2: Qualitative results on CrowdPose. Each image (left to right) shows DEKR predictions and our PINet predictions. Due to occlusions, DEKR often misses the occluded persons which are recovered by PINet.