# OpenReview forum: "Robust Pose Estimation in Crowded Scenes with Direct Pose-Level Inference"
_NeurIPS.cc/2021/Conference — NeurIPS 2021 Poster_

### Official Review · Reviewer_umsY · 2021-07-16

**Rating:** 7
**Confidence:** 3

**Summary:**

This paper proposes a method for robustly estimating human poses in crowded scenes. The main idea is to first detect visible body parts, then estimate human poses from the body parts. Based on these estimated human poses, refinement and fusion are performed. Experiments on OCHuman and CrowdPose show the effectiveness of the proposed method.

**Limitations And Societal Impact:**

It doesn't seem to me that there's a potential negative societal impact of this work.

**Main Review:**

Strengths:
- The paper is very well-written and the figures are carefully designed to clearly present the idea. It is a pleasure to read the paper.
- The idea is novel: it starts from part detection instead of purely top-down or bottom-up like most previous methods.
- One of the modules (pose fusion) in the proposed method is general and can be applied to other pose estimation frameworks.
- Experiments are well designed and the results are promising.

Overall I think the paper is interesting to the audiences in NeurIPS. Some comments rather than weaknesses:
- It would be very interesting to know how accurate are the poses predicted from different parts. For example, how accurate are the poses predicted from the head alone?
- How are the different parts contributing to the final results? Ablation studies can be done to see what the results are if some parts are ablated.
- Would it be better if different pose refinement networks are learned for different parts? It might be useful, but it is also possible that the effect is very limited.

**Time Spent Reviewing:**

4

---

> ### Author Response · Authors · 2021-08-10
> **Author Response to Reviewer umsY**
>
> Thanks for your constructive comments. We clarify the questions as follows.
> - - -
> Q1: The accuracy of pose predicted from different parts. Ablation study on different parts.
>
> A1: Thanks! We conducted the suggested ablation study on different parts and report the performance in the following table (U, M and B denote the upper, middle and bottom part, respectively). We can observe that upper part (head) achieves the best performance (57.0%). This is consistent with our motivation that, head is more likely to be visible and visible parts can better distinguish different persons. We also tested different part combinations to verify their performance. The results show all parts contribute to the final performance, i.e., combining three parts leads to the best performance. We will add discussions to the updated manuscript.
>
> | Part | U    | M    | B    | U&M  | U&B  | M&B  | U&M&B |
> | ---- | ---- | ---- | ---- | ---- | ---- | ---- | ----- |
> | AP   | 57.0 | 56.6 | 52.3 | 59.4 | 59.2 | 57.2 | 59.8  |
> - - -
> Q2: Using different pose refinement network for different parts.
>
> A2: Thanks! We conducted the suggested experiment of using different PR module for each part, and the performance on OCHuman is 58.6%, lower than the original 59.8%. This could be because that using different PR module makes the optimization more difficult, i.e., increases the number of parameters to be learned and decreases the number of training samples for each PR module.

---

> > ### Comment · Reviewer_umsY · 2021-09-03
> > **Final Review**
> >
> > Thanks for the feedback! Overall I still think this paper is interesting to the vision community and will keep the original rating.

---

### Official Review · Reviewer_1ttQ · 2021-07-16

**Rating:** 5
**Confidence:** 4

**Summary:**

 This paper presents a one-stage pose estimation approach that applies PPG to infer multiple poses from multiple body parts (head, upper body, and lower body). These coarse poses are then refined by a GCN module. Finally, the Pose Fusion module is applied to get the final result.


**Ethical Concerns:**

No ethical issues found.

**Limitations And Societal Impact:**

The limitations are not adequately stated. Especially, why the performance on COCO is lower than other approaches.

**Main Review:**

Pros:
- The overall idea of the paper is easy to follow.
- The paper reads smooth.
 - The proposed method achieves the state-of-the-art performance on two recently proposed datasets, i.e. crowdpose and ochuman datasets.
- The ablation study clearly shows the improvement of different modules.

Cons:
- The novelty of this paper is limited. Using graph networks to refine the initial pose is not new. [Ref1] construct dynamic graphs to tolerate large variations in the human pose. And [21] propose an Image-Guided Progressive GCN to estimate the invisible joints to deal with the occlusions. [Ref2] propose a two-stage graph-based method that adopts a graph pose refinement module to model the human structure. Line58 “this is an original contribution on the Direct Pose-Level Inference (DPLI) for pose estimation in crowded scenarios, which differs from the previous top-down and bottom-up pipelines.” is over-claimed. As the GCN is simply applied to each initial pose for pose refinement, there is no much difference, and cannot be considered novel.
[Ref1]  Z. Qiu, K. Qiu, J. Fu, and D. Fu, “Dgcn: Dynamic graph convolutional network for efficient multi-person pose estimation,” National Conference on Artificial Intelligence, 2020.
[Ref2] J. Wang, X. Long, Y. Gao, E. Ding, and S. Wen, “Graph-pcnn: Two stage human pose estimation with graph pose refinement,” In European Conference on Computer Vision. Springer, 2020.
- As stated above, some related works about GCN are missing [Ref1, Ref2]. Line86-87, detailed comparisons with previous works [15,32,14,27] are required. As most of them follow a similar one-stage pose estimation paradigm.
-The proposed approach does not achieve the state-of-the-art performance on COCO (as reported in supplementary, mAP 67.2 this paper vs 68.0 DEKR). COCO and MPII are the most popular and standard benchmarks for evaluating human pose estimation, while OCHuman is a recently proposed dataset and there are only a few officially reported results. The readers will more care about the results on COCO and MPII.
-The limitations are not adequately stated. Especially, why the performance on COCO is lower than other approaches.
-If I understand correctly, the Pose Fusion module is a post-processing algorithm, which makes the whole pipeline not fully end-to-end trainable. The matching/fusion loss (or the loss of the final output pose) cannot be back-propagated to the backbone network.
-Please also report the runtime performance (inference speed).

Minor:
-Why do you highlight AP^H in red in Table4?

**Time Spent Reviewing:**

4 hours

---

> ### Author Response · Authors · 2021-08-10
> **Author Response to Reviewer 1ttQ**
>
> Thanks for your constructive comments. We clarify the questions as follows.
> - - -
> Q1: The novelty of the proposed methods.
>
> A1: Thanks! We agree that graph network not new in pose estimation. Compared with those works, our method is different in several aspects.
> 1) we adopt a different dynamic graph. [Ref1] constructs the dynamic graph by sampling a Bernoulli distribution on a soft adjacency matrix. Our graph is adaptively learnt by gradient descent during the model training. [21] and [Ref2] adopt a static graph, thus cannot dynamically adjust the keypoint connections to refine poses. In Table 1(c) we show that our dynamic graph is better than such static graph. We will add [Ref1] and [Ref2] to the manuscript.
> 2) only utilizing the graph cannot fully handle occlusion. For example, [21] and [Ref1] achieve relatively low performance on crowded scene (29.1% on OCHuman and 59.1% on CrowdPose). [Ref2] is based on the top-down pipeline and cannot address the missed person detection issue in crowded scenes (line 23~30). Our method effectively addresses those issues with the PPG and PF, e.g., utilizes visible parts to distinguish different persons and infer the complete person pose. Experimental results show that our method obtains superior pose estimation performance on crowded scenes.
> - - -
>
> Q2: Comparison with previous one-stage methods.
>
> A2: Thanks! Previous one-stage methods mainly detect person center point and regress keypoint offsets, and cannot work in crowded scenes well. First, the center point cannot effectively distinguish different persons in crowded scenes, because they can be occluded as verified in Figure 3. Our work relies on visible parts to distinguish persons, thus is more robust to occlusion. Both Figure 3 and Table 2 demonstrates the superiority of our method. Second, previous works also suffer from the difficulty of long distance regression problem. Our Pose Fusion (PF) selects the most confident keypoints in poses generated from multiple parts, and effectively alleviate this problem.
> - - -
> Q3: The performance of COCO and MPII.
>
> A3: Thanks! We agree that COCO and MPII are most popular multi-person pose estimation benchmarks. This paper focuses on crowded scenes pose estimation [a, b] which is more challenging because persons in the image are highly overlapped (line 23-30). In [b] the authors summarize existing benchmarks and show that COCO and MPII contain less crowded scenes. For instance, there is no image that has an average IoU larger than 0.5 on MPII. Therefore, we test our methods on CrowdPose and OCHuman, which are more suited and popular to serve as test sets for crowded scenes pose estimation task.
>
> | Dataset   | Total  | IoU<0.3 |  IoU>0.5  |
> | --------- | :----- | :------ | :-------- |
> | MPII      | 24,987 | 24,987(100%)  |     0(0%)     |
> | COCO      | 118,287 | 111,783(95%) | 1,209(1%)  |
> | CrowdPose | 20,000  | 11,294(56%)  | 2,909(15%) |
> | OCHuman   |  4,731  |  1,467(32%)   | 3,244(68%) |
>
> In fact, our method achieves comparable performance to previous works on COCO. For instance, we obtain 67.2% AP on COCO val set, which is comparable to HigherHRNet (67.1), HGG(60.4) and PersonLab (66.5). DEKR is published at CVPR2021 in June, and can be regraded as a contemporary work. DEKR and our work have difference focuses, i.e., improving regression-based pose estimation in general scenes and improving pose estimation in crowded scenes, respectively. Our method performs worse than DEKR on non-crowded COCO, but substantially outperforms it on crowded scenes, e.g., 68.9% vs. 65.7% on CrowdPose. Our method also has different methodologies with DEKR, technologies such as separate regression and adaptive convolution proposed in DEKR can also be applied to our method.
>
> [a] J. Li, C. Wang, H. Zhu, Y. Mao, H.-S. Fang, and C. Lu. Crowdpose: Efficient crowded scenes pose estimation and a new benchmark. In CVPR, 2019.
>
> [b] L. Qiu, X. Zhang, Y. Li, G. Li, X. Wu, Z. Xiong, X. Han, and S. Cui. Peeking into occluded joints: A novel framework for crowd pose estimation. In ECCV, 2020.
> - - -
> Q4: Pose Fusion is not differentiable and the whole pipeline is not fully end-to-end trainable.
>
> A4: Yes, Pose Fusion is a non-differentiable operation. It is a post-processing operation only used during test, which is similar to NMS. During training we train the backbone, PPG and PR in an end-to-end manner. We will clarify this in updated manuscript to avoid confusion.
> - - -
> Q5: Runtime performance.
>
> A5: Thanks! We test our model on a GTX 1080Ti and it runs at 14.7 FPS. We also list the runtime of some other methods in the following table. We will add more discussions about running speed and inference time in the updated manuscript.
>
> | Method | HigherHRNet | OpenPose | Ours (PINet) |
> | ------ | :---------: | :------: | :----------: |
> | FPS    |     7.8     |   13.5   |     14.7     |
> - - -
> Q6: $AP^H$ in Table 4.
>
> A6: CrowdPose divides test set into three parts according to the CrowdIndex, i.e., easy, medium, and hard sets, respectively. The hard test set contains more crowded scenes than the other two. We thus highlight the $AP^H$ to show the improvement of our method on the most challenging test set. For instance, our method is higher than HigherHRNet by 3% on the easier test set ($AP^E$) and 3.6% on the hardest test set ($AP^H$).

---

> > ### Comment · Reviewer_1ttQ · 2021-08-24
> > **Feedback to the author response**
> >
> > Thanks for your reply. I have carefully read all the discussions between other reviewers and the authors.
> >
> > I agree with Reviewer wMdR, that this paper presents a well-engineered human pose estimation system tuned for crowded scenes. However, the novelty of this paper is not sufficient enough for publication in NeurIPS.
> >
> > I also agree with Reviewr AzWE that training on OChuman validation set is not a standard setting. According to the author of OCHuman, “OCHuman dataset is designed for validation and testing … it is better to use general datasets such as COCO as a training set, then test the robustness … to occlusion using this dataset, rather than performing training on only occluded cases.”
> >
> > Experiments are not very convincing.
> > 1.	The comparisons in Table 1(b) are not fair. The results of “Center” is obtained without PPG & PR (see Table 1(a)), but the results of “Part-large” is obtained using PPG & PR & PF.
> > 2.	In the reply to Reviewer M2eL A4, why the results of Occlusion-free is worse than those of Occlusion. This implies that the proposed method might perform good in crowded scenes, but might lead to performance drop in Occlusion-free scenes due to duplicated poses.

---

> > > ### Author Response · Authors · 2021-08-25
> > > **Response to Reviewer 1ttQ**
> > >
> > > Thanks for your comments.
> > >
> > > To the best of our knowledge, this work is an original contribution on the direct pose-level inference, which is free of bounding box detection and keypoint grouping commonly required by previous top-down and bottom-up pipelines. It relies on visible person parts to distinguish different persons and infer the complete person pose, thus avoids issues of missed person detection and missed keypoint localization. As illustrated by our experiments, this framework is easy to implement and works well.
> > >
> > > - - -
> > > Training on OCHuman val set is not new in our work. This setting is also used in another crowded scenes pose estimation work [21]. We follow this setting in [21] and report the performance of several methods in Table 2. We also report the performance on OCHuman by training on COCO in the supplementary material (illustrated in the following table). Under this commonly used setting, our method still achieves the best performance.
> > >
> > > | Method (trained on COCO) | OCHuman val | OCHuman test |
> > > | :---------------- | :---------: | :----------: |
> > > | AE[12]     |    32.1     |     29.5     |
> > > | HGG[7]    |    35.6     |     34.8     |
> > > | DEKR[4]   |    34.5     |     33.2     |
> > > | Ours   |    38.3     |     37.2     |
> > >
> > > - - -
> > > Thanks for pointing out this. The performance of part-large without PR&PF is 53.1 (listed in the second row ofof Table 1(a)). Under the same setting, our method also performs performs better than other partition strategies in Table 1(b), e.g., center and part-small.
> > > - - -
> > >
> > > Note that, occlusion and occlusion-free subsets are different in both contained images and dataset size, i.e., 706 vs. 1525 images. Different test sets make their reported performance not directly comparable. We have tested other methods like baseline (SPM[15]) and DEKR[4] (trained on OCHuman val set or COCO train set) on these two subsets. We observed that they also get lower performance on occlusion-free subset than on the occluded one. We hence would like to note that, lower performance on the occlusion-free subset cannot imply that our method may lead to performance drop in occlusion-free scenes. Under a more reasonable setting of comparing to baselines, our method improves the performance on both occlusion-free and occlusion subsets.
> > >
> > > | Method      | occlusion-free | occlusion |
> > > | :---------- | :------------: | :-------: |
> > > | baseline   |      30.8      |   55.6    |
> > > | DEKR[4]        |      34.5      |   60.3    |
> > > | Ours        |      38.0      |   69.7    |
> > > | DEKR (coco) |      21.6      |   39.2    |
> > > | Ours (coco) |      23.0      |   43.7    |

---

> > > > ### Comment · Reviewer_1ttQ · 2021-09-03
> > > > **Final rating**
> > > >
> > > > Thanks for the author feedback. The results have answered most of my questions. I will upgrade the final rating.

---

### Official Review · Reviewer_ciZS · 2021-07-16

**Rating:** 6
**Confidence:** 4

**Summary:**

The paper introduces a novel method that is able to extract human pose information in crowded scenes. The authors present a bottom-up  method that does not require detection or keypoint grouping. Their method infers initial pose cues from body part. The initial part cues are refined and fused to get the pose cues for the occluded people. They evaluate their work on 2 datasets.

**Limitations And Societal Impact:**

The authors have talked about limitations in Section 5 and have mentioned that there are not negative societal impacts.

**Main Review:**

Hereafter, I will share the strengths and weaknesses of the paper:

[Strengths]
- The problem is relevant.
- Good ablation study.

[Weaknesses]
-	The statement in the intro about bottom up methods is not necessarily true (Line 28). Bottom-up methods do have a receptive fields that can infer from all the information in the scene and can still predict invisible keypoints.
-	Several parts of the methodology are not clear.
		- PPG outputs a complete pose relative to every part’s center. Thus O_{up} should contain the offset for every keypoint with respect to the center of the upper part. In Eq.2 of the supplementary material, it seems that O_{up} is trained to output the offset for the keypoints that are not farther than a distance \textit{r) to the center of corresponding part. How are the groundtruths actually built? If it is the latter, how can the network parts responsible for each part predict all the keypoints of the pose.
		- Line 179, what did the authors mean by saying that the fully connected layers predict the ground-truth in addition to the offsets?
		- Is \delta P_{j} a single offset for the center of that part or it contains distinct offsets for every keypoint?
		- In Section 3.3, how is G built using the human skeleton? It is better to describe the size and elements of G. Also, add the dimensions of G,X, and W to better understand what DGCN is doing.
-	Experiment can be improved:
		- For instance, the bottom-up method [9] has reported results on crowdpose dataset outperforming all methods in Table 4 with a ResNet-50 (including the paper one). It will be nice to include it in the tables
		- It will be nice to evaluate the performance of their method on the standard MS coco dataset to see if there is a drop in performance in easy (non occluded) settings.
		- No study of inference time. Since this is a pose estimation method that is direct and does not require detection or keypoint grouping, it is worth to compare its inference speed to previous top-down and bottom-up pose estimation method.
		- Can we visualize G, the dynamic graph, as it changes through DGCN? It might give an insight on what the network used to predict keypoints, especially the invisible ones.

[Minor comments]
- 	In Algorithm 1 line 8 in Suppl Material, did the authors mean Eq 11 instead of Eq.4?
-	Fig1 and Fig2 in supplementary are the same
-	Spelling Mistake line 93: It it requires…
-	What does ‘… updated as model parameters’ mean in line 176
-	Do the authors mean Equation 7 in line 212?


**Time Spent Reviewing:**

3

---

> ### Author Response · Authors · 2021-08-10
> **Author Response to Reviewer ciZS**
>
> Thanks for your constructive comments. We clarify the questions as follows.
> - - -
> Q1: Statement is not accurate. Bottom-up methods can infer invisible keypoints.
>
> A1: Thanks. We agree that bottom-up methods have the ability of inferring invisible keypoints. However, there still exist some issues for typical bottom-up methods (detect-and-group). Invisible keypoints inferred from heatmap tend to have relatively low confidence score. On HigherHRNet, we tested a well-trained model and find that, the average confidence score of invisible keypoints is about 0.02, while the score of visible keypoints is about 0.34. Too low confidence score means that invisible keypoints are likely to be discarded during testing. Moreover, for two overlapped keypoints, only one could be detected. Those issues make most of existing keypoint grouping operations not robust to occlusion and missed detected keypoints. We will update the discussions about bottom-up methods in revised manuscript.
> - - -
> Q2.1: The ground truth offset map construction.
>
> A2.1: We apologize for the confusion. Suppose the part center $p=(x,y)$ and the person pose $\mathcal P=\\{p_k\\}_{k=1}^n$, offsets  at $(x,y)$ are computed as $\text{O}(x,y)=\\{p_1-p,…,p_k-p,..., p_n-p\\}$. The $r$ denotes that we generate offsets at part center $(x, y)$, and its neighboring position ($\mathcal N(p)$) with radius $r$. $r$ is set to 2 in all experiments.
> - - -
> Q2.2: Line 179, what did the authors mean by saying that the fully connected layers predict the ground-truth in addition to the offsets?
>
> A2.2: The output of fully connected layers is the difference between initial generated pose $\mathcal P_j$ and the ground truth $\mathcal P_j^*$. We will modify this saying in the updated manuscript.
> - - -
> Q2.3: Is $\Delta \mathcal P_j$ a single offset for the center of that part or it contains distinct offsets for every keypoint?
>
> A2.3: $\Delta \mathcal P_j$ contains distinct offsets for every keypoint. We will clarify this in the manuscript.
> - - -
> Q2.4: Details about DGCN parameters.
>
> Q2.4: The dimension of $G$ is $n * n$, where n denotes the number of keypoints for one person. $X$ is a $N * D$ size matrix, where $N$ is the number of sampled poses for PR, $D$ is the feature dimension, which is set to 480 in all experiments. $W$ is a $D * D$ size matrix, which is the parameter of DGCN. $G$ is an adjacency matrix that reflects the connection of keypoints. If the i-th keypoint and j-th keypoint are connected, then $G[i][j] = 1$. We normalize $G$ following [a].
>
> [a] Thomas N Kipf and Max Welling. Semi-supervised classification with graph convolutional networks. ICLR, 2017
> - - -
> Q3.1: Missing related works.
>
> A3.1: Thanks for suggestion. Those works will be cited and discussed.
> - - -
> Q3.2: Performance on COCO.
>
> A3.2: We provide the performance comparison on COCO in supplementary material. Our method achieves comparable performance with other methods.
> - - -
> Q3.3: Inference time.
>
> A3.3: Thanks! We test our model on a GTX 1080Ti and it runs at 14.7 FPS. We also list the runtime of some other methods in the following table. We will add more discussions about running speed and inference time in the updated manuscript.
>
> | Method | HigherHRNet | OpenPose | Ours (PINet) |
> | ------ | :---------: | :------: | :----------: |
> | FPS    |     7.8     |   13.5   |     14.7     |
> - - -
> Q3.4: Visualization of the dynamic graph.
>
> A3.4: Figure 2 illustrates some examples of the dynamic graph generated during training. It can have be observed that, the training procedure adds some new connections denoted by red solid lies, and removes some connections denoted by red dashed lines. Descriptions will be added to the caption of Figure 2.
> - - -
> Q4: Some typos and what does ‘… updated as model parameters’ mean in line 176.
>
> A4: Thanks for pointing out those typos. We will carefully check and correct typos. Line 176 means that we set the dynamic graph as the initial parameters of the model, which are hence updated during the training procedure.

---

> > ### Comment · Reviewer_ciZS · 2021-09-02
> > **Final review**
> >
> > Thank you for all your answers. I will keep my initial rating. All the best.

---

### Official Review · Reviewer_8zdn · 2021-07-17

**Rating:** 7
**Confidence:** 5

**Summary:**

In this paper the authors study the problem of the multi-person pose estimation in crowded scenes. They propose a new method which directly infers poses by leveraging both body part cues and pose structure prior. In the experiments, the authors could show some promising results on crowded scenes.

**Limitations And Societal Impact:**

Overall this paper studies an under-explored problem of pose estimation for crowded people. Overall I feel this paper is well prepared and I incline to accept this paper. The authors are encouraged to respond my questions in the rebuttal.

**Main Review:**

Overall this paper is well written and easy to follow. The author studies a challenging problem of pose estimation for crowded people and the proposed approach works well compared with existing approaches.
- The idea of direct pose inference is interesting and could be quite efficient by getting rid of person detection in the pipeline.
- The proposed framework consists of part-based pose generation, pose refinement and pose fusion. Pose refinement module and pose fusion module are adapted from existing methods and method novelty might be incremental.
- The experiments are conducted on three benchmarks: OCHuman, CrowdPose and COCO. The comparisons with state-of-the-art are quite convincing.

I have a few questions and comments for this paper.
- How would the proposed approach handle keypoints that are out of view? Would it be possible to further adapt the part-based pose generation module to different modes (e.g., head only, upper-body only) instead of always predicting the whole body?
- The design for body partition seems a little bit ad-hoc. Why the authors discard arms for part-large in the body partition illustrated in Fig. 3? If this is determined empirically, could you explain why a variant of this strategy which include arms in the torso part perform worse? Is this design optimal for all the datasets?
- In Fig.4, it is hard for me to tell whether PPG+PR or PPG+PR+PF performs better. Could you provide more example results to illustrate under what scenarios do PF improve?
- The experiments on COCO are quite important to validate whether the proposed approach obtains better performance for occluded people poses with the tradeoff of regular poses. The authors should consider moving this experiment from supplementary to the main text in the final version.




**Time Spent Reviewing:**

2

---

> ### Author Response · Authors · 2021-08-10
> **Author Response to Reviewer 8zdn**
>
> Thanks for your constructive comments. We clarify the questions as follows.
> - - -
> Q1: How to handle keypoints that are out of view?
>
> A1: For keypoints that are out of view, the PPG module first makes a global inference to estimate a coarse location of each keypoints for each person. Then PR refines the location of invisible keypoints by incorporating context cues and person pose structure prior.
> - - -
> Q2: Predicting specific keypoints, not whole body in PPG.
>
> A2: Predicting specific keypoints for each part (e.g., head part only predicts the keypoints on the head) would be better to avoid long distance regression problem, but can introduce several new issues. 1) it is non-trivial to group these parts into a full person body. In other words, this strategy also confronts the pose grouping issue of bottom-up methods in crowded scenes. 2) once some parts of one person are occluded or not detected, it is hard to recover them from other visible parts because they only detect partial keypoints. We hence think predicting the whole body is a more reasonable solution.
> - - -
> Q3: The design of body partition seems a little bit ad-hoc, and why remove arms from torso part?
>
> A3: Thanks! Our body partition strategies are carefully designed and are not ad-hoc. Previous works [a, b] present typical body partitions, such as the widely used six parts (the fourth one in Figure 3). We have studied this setting, a coarser partition (three parts, the third one in Figure 3) and a finer partition (13 keypoints, the fifth one in Figure 3) in Table 1(b).
>
> The three parts partition (part-large) divides a person body into upper, middle, and bottom parts. It is sufficient to locate the middle part center with torso area, therefore including arms is unnecessary. Moreover, arms commonly present larger degree of freedom than torso area, e.g., arms may appear at the upper or bottom part. Therefore, introducing arms may have a negative impact on locating the middle part. We also conduct experiments on two datasets and report the performance in the following table. We can observe that introducing arms into the middle part is not helpful. We hence exclude it from the middle part.
>
> | Dataset      | OCHuman | CrowdPose |
> | :----------- | :-----: | :-------: |
> | with arms    |  58.9   |   68.4    |
> | without arms |  59.8   |   68.9    |
>
> [a] R. Pytel, O. S. Kayhan, and J. C. van Gemert. Tilting at windmills: Data augmentation for deep pose estimation does not help with occlusions. ICPR, 2020.
>
> [b] Li, S. Zhang, Q. Tian, M. Wang, and W. Gao. Pose-guided representation learning for person re-identification. IEEE transactions on pattern analysis and machine intelligence, 2019.
> - - -
> Q4: It is hard to observe the improvement of PF in Fig.4.
>
> A4: Thanks for pointing out. In the top-left example of Figure 4, PF improves the location of right knee of the person with golden hair. In the top-right example, PF improves the left knee and ankle of the little girl. We will add description and more obvious examples to the updated manuscript.
> - - -
> Q5: Including COCO experiments to the manuscript.
>
> A5: Thanks for your suggestion, we will add this to the updated manuscript.
> - - -

---

### Official Review · Reviewer_M2eL · 2021-07-17

**Rating:** 6
**Confidence:** 4

**Summary:**

This paper proposes a three-step approach to conduct multi-person pose estimation with occlusion. Firstly, their framework starts with a part-based pose generation (PPG) module aiming at providing a set of body pose candidates, where the initial prediction could be less considering the holistic body pose information. Then, a dynamic graph convolutional network (DGCN) is applied to incorporate the holistic pose structure information for the whole set's pose refinement (PR). Finally, a pose fusion (PF) module is proposed to merge and ensemble from the multiple outputs of the previous two steps.

The paper builds a sophisticated framework to conduct the key point detection task, where each of the module is with careful design. The contribution of this paper is combining the pose regression (or termed PPG in the paper), graph based pose refinement and pose fusion together to achieve some good performance, where the architecture design and training could be non-trivial. There are some informative discussions such as, which body partition would be better for the optimization (Fig. 3), the importance of each of their proposed module (Table 1). Further, the consistent advantageous results achieved on two pose estimation datasets, OCHuman and CrowdPose.

**Limitations And Societal Impact:**

Yes, for the human images they used in the paper, the authors all masked out the face area for the privacy consideration.

**Main Review:**

* Originality:

The overall framework is consists of three major components, PPG, PR and PF. Such combination is not seen in any of the related work. However, if looking into each of the component itself, the part-based pose generation essentially is a pose regression problem to regress both the center heat map P and the offset heat map O, which is a general representation as appeared in [15]. PR leverages dynamic graph convolutional network to conduct the key point refinement, where similar work [21] has applied the GCN (progressive GCN) in the framework to solve the same problem. The authors provide some discussion regarding their DGCN difference from the GCN in [21]. It is not very clear how their proposed G is dynamic, and how such dynamic graph is determined based on each pose map input. Finally, the PF step leverages the OKS proposed from [17] as the rule to determine the duplicate pose detections. Overall, I think the originality is okay but not strong enough.

* quality:

The paper provides a good structure with sufficient problem and background introduction. The connection of their idea to the related work is also well addressed. The figures are clear, meaningful and easy for the readers to understand. There are sufficient implementation details for the readers to try and replicate their method. The experiment is towards thorough to cover most of their proposed modules, where consistent advantageous performance is achieved. Overall, I think the presentation quality of the paper is good and above average.

* clarity:

For large part of their method, it is clear. However, there are also some other places that the authors need to take care of.

  ** It is not straight-forward why the proposed PPG can deal with occlusion. As generally, PPG is based on the appearance input to regress the key point heat map, each part detects its own key points. There is no extra setting for the key points to indicate its occlusion likelihood.

Further, the occlusion can be inferred because of the holistic pose structure, where the GCN based pose refinement is to deal with the occlusion. It is also evidence in [21], where it states in Sec. 3.2 "estimated position of invisible joints from the base module is sometimes far from their correct locations and this makes it challenging to directly regress their displacements Therefore, we design an intuitive coarse-to-fine learning mechanism in the coordinate-based module, which builds a progressive GCN architecture and leverages the performance steadily by enforcing multi-scale image features in a progressive manner".

If the above is correct, the dealing of occlusion is from GCN, not from PPG. It'll require the authors to carefully think about their design and rephrase their motivation towards each of the proposed module.

  ** Line 207, r_k is obtained from H: extra score map, what is the difference between r_k and r_p?

The authors mention to obtain r_k from ''heat map'' H. However, usually, the heat map is mostly regarding to the pose heat map P. It is not clear why there are several ''heat maps''. And why need this extra heat map H to infer the r_k for the confidence score? Would r_p be sufficient to tell the confidence here?

  ** The method emphasis the capability of handling occlusion. There is dataset OCHuman which is the occluded human body dataset. From the experiments, there is average precision (AP) to indicate the performance, i.e., Table 1 (a) to indicate the effectiveness of each module. However, it is not clear which part is to handle the occlusion. It would be good to separate the testing set into occlusion-free subsets and occlusion subsets, which at least would further tell under occlusion heavy situation, what module is critical in handling it.

Overall, I think the clarify of the paper needs further improvement, where there are several key parts need to be clarified.

* Significance

The overall framework provides a reasonable solution to achieve the multi-body pose estimation. There are several comments that need the authors to further clarify.

**  for part-based pose generation (PPG), according to the loss design, (Eqn. 7 and 8), the optimization are sort of deterministic, whereas it is not likely to have many optimal P and O.

It needs further clarification under what constraint with the objective of Eqn. 7 and 8, the optimization will lead to “duplicate poses and low-quality poses from invisible body parts”.

** consider the order of pose refinement (PR) and pose fusion (PF), given that PPG is likely to generate multiple pose hypothesis, what would be the better order, i.e., PPG->PF->PR or PPG-PR-PF?

it is likely from the inaccurate PPG, pose refinement further lead to noise amplified results, where the pose fusion is hard to vote the scattered key points.  While after PPG, firstly conduct pose fusion would provide a relatively more confident intermediate pose heat maps. The further pose refinement (PR) would refine to reduce the local distortions.

It would be good to demonstrate the effectiveness of the order of conducting PR and PF.

** it lacks the ablation with or without the M for the loss. Usually the human body crop is already with little background, which is needed to verify how different by adding this background mask for the loss design.

** it lacks cross dataset evaluation. For the shared joints, it would be good to see the performance. As network is likely to be overfitting to the specific dataset. Further, if the distribution of the training set (validation set) and the testing set is exactly the same, it is actually not convincing by showing high performance on the testing set.

-minor:
* Would non-maximum suppression (NMS) achieving similar effect as the proposed PF? If not, please provide some analysis or experimental results for that.

* Inconsistent naming of the proposed method, PINet appeared in most of the places, where DPLI appeared in most of the introduction part, and in Figure 1 caption. It needs to be unified.

By combining all the above aspects together, I think the paper provides a relatively original solution for a mature field of human pose estimation. However, there are several parts that need the authors to carefully identify, such as the occlusion handling, the design of PPG->PR->PF, whether the PPG will generate many P and O for the later PF stage and so on.  I would like to further hear from the authors for their analysis.


**Time Spent Reviewing:**

6hrs

---

> ### Author Response · Authors · 2021-08-10
> **Author Response to Reviewer M2eL**
>
> Thanks for your constructive comments. We clarify the questions as follows.
> - - -
> Q1: Originality of the proposed method.
>
> A1-1: PPG is different from [15] in that it relies on visible parts to distinguish different persons in crowded scenes, rather than using the person center point. As states in Section 3.2 of the manuscript, the part design is critical to handle the crowded scenes and occlusion. Table 1 (a) and (b) show the superiority of PPG.
>
> A1-2: Thanks! We agree that PR shares similarities with [21], in that both of them use GCN. But there are some differences: 1) different graph: PR adopts dynamic graph to refine keypoint features, which is updated with model during training. This property allows it to increase or decrease keypoint connections to extract better features. [21] use a static graph. Table 1(c) shows that dynamic graph performs better than static graph. 2) different implementation: the GCN in [21] takes keypoint coordinates as input, and requires extra module to extract context cues. PR uses GCN as an aggregation process to extract better person global feature. PR does not require extra module, thus is potentially more straightforward and efficient.
>
> A1-3: The OKS criterion is widely used to find duplicate pose detections in previous work and is not novel. The core component of PF is to fuse these duplicate poses and remove low quality ones. Our experiments show that this strategy is effective in addressing the long distance regression issue.
> - - -
> Q2: How PPG deal with occlusion?
>
> A2: Occlusion in multiple person scenarios causes two issues: 1) missed person detection as shown in Figure.1(a), and 2) missed keypoint localization as shown in Figure.1(b). Previous work, e.g., OPEC-Net [21] only focuses on the second issue and ignores the first one. Our method effectively addresses those two issues with a uniform framework. To deal with the first issue, our PPG uses visible person parts to differentiate persons in crowded scenes, because visible parts are easier to detect than invisible parts. To address the second issue, PPG regresses the complete pose of a person based on each of his/her detected part. This strategy ensures: 1) every person could be differentiated by one or multiple parts, and 2) every person has as least one complete pose, thus effectively address those two issues caused by occlusion.
>
> The PR module is proposed to deal with the inaccurate keypoint localization by PPG. It cannot fully handle occlusion. We conduct another experiment to remove PPG and keep GCN, and it obtains 52.1% AP, lower than the 59.8% achieved with PPG. This indicates that both PPG and GCN in PR are important for handling occlusion.
> - - -
> Q3: What is the difference between $r_k$ and $r_p$? the effect of extra score map.
>
> A3: PPG generates the person pose by detecting person part center and regressing offsets of keypoints. $r_p$ denotes the confidence of a generated pose $\mathcal P$. Note that $\mathcal P$ is composed with multiple keypoints. To further denote the confidence of each estimated keypoint, we introduce $r_k$, which denotes the confidence of the k-th keypoint in pose $\mathcal P$. In other words, $r_p$ is obtained from the part center heatmap, while $r_k$ is obtained from keypoint heatmap (extra score map).
> - - -
> Q4: Splitting testing set into occlusion-free and occlusion subsets to investigate which modules are critical in handling occlusion.
>
> A4: Thanks for your suggestion. We split OCHuman test set into two parts: occlusion-free subset (IoU <=0.1, 706 images) and occlusion subset (IoU>=0.3, 1525 images) and report the performance in the following table. It is clear that, our method brings more substantial performance gains on the occlusion subset, e. g., improving the performance from 55.6% to 69.7%. We also observe that, both PPG and PR are critical in handling occlusion, i.e., they bring 6.4% and 6.9% improvements, respectively on the occlusion subset.
>
> | PPG  |  PR  |  PF  | Occlusion-free | Occlusion | ALL  |
> | :--: | :--: | :--: | :------------: | :-------: | :--: |
> |      |      |      |      30.8      |   55.6    | 47.6 |
> |  o   |      |      |      33.2      |   62.0    | 53.1 |
> |  o   |  o   |      |      36.8      |   68.9    | 58.8 |
> |  o   |      |  o   |      45.0      |   65.0    | 55.8 |
> |  o   |  o   |  o   |      38.0      |   69.7    | 59.8 |
> - - -
> Q5: The reason for duplicate poses and low-quality poses.
>
> A5: We divide each person into three parts, thus there will be three $\textbf{P}$ and $\textbf{O}$, and each $\textbf{P}$ and $\textbf{O}$ correspond to a specific part. The optimal $\textbf{P}$ and $\textbf{O}$ of each part are deterministic. For j-th part, we can detect one person by finding the local maximum on $\textbf{P}_j$ and sampling $\textbf{O}_j$ to generate pose. Conducting this procedure on three parts may generate 3 duplicate poses at most. Since some parts may be invisible due to occlusion, poses generated from those parts are inaccurate. Therefore, PPG could generate duplicate and low quality poses.
> - - -
> Q6: The order of PPG, PR and PF.
>
> A6: It is interesting to exchange the order of PR and PF. PF is a non-differentiable post-processing operation. We cannot implement PPG->PF->PR pipeline under current design, because PPG and PR cannot be optimized end-to-end in the this pipeline. Moreover, running PF before PR may cause the wrong matching in Eq.(11) as it requires accurate pose estimation results. In other word, PPG->PR generate more accurate pose than only applying PPG, making PPG->PR->PF a more reasonable choice.
> - - -
> Q7: Ablation study on the mask in heatmap loss computation.
>
> A7: The input of our method is the original image, not cropped person patch. Therefore, our input contains lots of backgrounds, making it helpful to add a mask to down weight the loss of background. We conducted an experiment to remove the mask from heatmap loss, and the performance is 54.3%, lower than the 59.8% achieved with mask.
> - - -
> Q8: Cross dataset evaluation.
>
> A8: Thanks! It is interesting to do cross dataset evaluation for pose estimation. To the best of our knowledge, there is no pose estimation work that conducts such experiment. We do cross dataset evaluation on OCHuman and CrowdPose and report the performance in the following tables. We can observe performance drop under such setting. For instance, the model trained on CrowdPose achieves 68.9% on CrowdPose and 49.4% on OCHuman. This indicates that domain gap degrades the performance of pose estimation, which can be studied in future research. Under this setting, our method (the left value in the table) still performs better than a recent work DEKR (the right value in the table), indicating that our model is less prone to overfit.
>
> | Ours \| DEKR | OCHuman(train) | CrowdPose(train) |
> | :----------------- | :------------: | :--------------: |
> | OCHuman(test)      |  59.8 \| 52.1  |   49.4 \| 46.3   |
> | CrowdPose(test)    |  31.0 \| 21.3  |   68.9 \| 65.7   |
>
> - - -
> Q9: Does NMS achieves similar performance as PF?
>
> A9: NMS can be used to discard duplicate predictions. As discussed in Section 3.4 (line192-196), due to the difficulty of long distance regression, fusing duplicate predictions could performs better than discarding them. For instance, the pose regressed from upper part performs well on locating keypoints on head, while the pose inferred from the bottom part is more accurate in locating ankle and knee. Fusing those two poses generates more accurate pose estimation. We have verified this in experiments. Results of the first three rows in Table 1(a) are obtained by utilizing NMS as post-processing to remove duplicate poses. After replacing NMS with PF, the performance improves by 1.0~2.7% in AP. We hence conclude that PF is better than NMS.
> - - -
> Q10: Inconsistent naming of the proposed methods.
>
> A10: Thanks for pointing this out. DPLI denotes the pipeline that inferring person pose from visible person parts. PINet is an implementation of DPLI. We will clarify this in the updated manuscript.

---

> > ### Comment · Reviewer_M2eL · 2021-08-30
> > **Reply to Author Response**
> >
> > I appreciate the authors' efforts in clarifying each of the points. I think large portion of their replies clearly address my concern, i.e., the technical details as well as the further ablative study. There are several remaining points that I would like to discuss with the authors.
> >
> > - DGCN is not clear enough in the paper script, how the parameters are dynamically maintained?
> >
> > For GCN, there is the "G" indicating the adjacency among the parts. According to the authors, their "PR adopts dynamic graph to refine keypoint features". If such "G" is dynamically formed, it would be good to further explain in detail about such key step.
> >
> > There is some clarification reply to Reviewer ciZS Q2.4. But it does not indicate how to deal with those dynamically changed connection in "G".
> >
> > -  PPG to deal with occlusion
> >
> > The authors mentioned: "our PPG uses visible person parts to differentiate persons in crowded scenes". I believe here the authors will need to make one important information clear to the readers (it seems to lack at least text-wise, that's where originally I got confused): the detected parts (i.e. 3 part design, upper/middle/lower body) will indicate the person information, i.e., the PPG will return one set of upper/middle/lower body per person.
> >
> > In this way, the holistic information is utilized for those part detections. Otherwise, leaving the PPG only outputs the parts without holistic information, it seems hard to overcome the occlusion.
> >
> > - A4 results on occlusion-free and occlusion-heavy subsets
> >
> > Thanks for the further detailed report. It is very helpful. But it seems potentially opening up a new concern: it is counter-intuitive that occlusion-free performance is consistently inferior than the occlusion-heavy performance. Does it mean occlusion-free task is harder than occlusion-heavy pose parsing? Or it could be a side-evidence that the model is not generalized well. The reason could be from the training data distribution, i.e., the occlusion-free versus occlusion-heavy training samples are biased towards the latter. Please find out and bring a discussion towards it.
> >
> > There is some clarification reply to Reviewer 1ttQ. It is okay to see for other methods, they also appear with inferior performance with the occlusion-free subsets. However, it'll require some efforts to justify why performance on occlusion-free subset is poorer.
> >
> > Overall, I think the authors' reply clearly improves the paper quality and I hope all those can be incorporated into the paper script. If the further above points can be clarified, I am more than happy to raise my original rating.

---

> > > ### Author Response · Authors · 2021-08-30
> > > **Response to Reviewer M2eL**
> > >
> > > Thanks for your comments! We will carefully revise our paper according to the above discussion.
> > > - - -
> > > Q1: DGCN is not clear enough in the paper script, how the parameters are dynamically maintained?
> > >
> > > A1: Thanks for the comments! The graph "G" indicates a two-dimensional matrix reflecting the adjacency among keypoints. As shown in Eq. (9), DGCN refines keypoint features with keypoint relationship encoded in "G". “dynamic graph” means that the graph "G" is adaptively adjusted during training to optimize keypoint adjacency and minimize the loss in Eq. (10). The graph is adjusted by the backpropagation algorithm to seek better pose refinement result. During training, we first initialize a pre-defined "G" as a tensor in the model, then update it through the optimizer for each iteration. This procedure adaptively learns a graph for the training set, which is hence adopted for testing.
> > > - - -
> > >
> > > Q2: PPG to deal with occlusion.
> > >
> > > A2: Thanks for the suggestion! Yes, each detected part indicates a holistic person pose cue. This is achieved by first detecting the part center, then regressing the keypoint offsets with respect to this part center. The resulting part center and keypoint offsets are combined to infer a holistic person pose cue. For each person, PPG outputs three holistic person pose cues inferred by upper/middle/lower parts, respectively. PF hence fuses three holistic pose cues to get the final result. We will update the manuscript to clarify those descriptions.
> > > - - -
> > > Q3: A4 results on occlusion-free and occlusion-heavy subsets
> > >
> > > A3: Thanks for the comments. The occlusion-free subset is generated by: 1) first computing the mean IoU based on annotated persons in each image, 2) classifying images with low mean IoU into the occlusion-free subset. Note that, unannotated occluded persons are not counted for the computation of mean IoU. E.g., for an image containing two occluded persons, if one person is not annotated, then the IoU for this image would be zero. As a result, many images containing occluded persons are classified into the occlusion-free subset. We have checked the occlusion-free subset and found that those images degrade the performance. Moreover, unannotated persons lead to a larger number of false positives in AP computation (i.e., those detected persons can not be matched to ground truth because they are not annotated), thus also degrade the performance.

---

> > > > ### Comment · Reviewer_M2eL · 2021-09-03
> > > > **response to author feedback**
> > > >
> > > > Thanks for the further clarification. I have already upgraded my rating.

---

### Official Review · Reviewer_wMdR · 2021-07-17

**Rating:** 5
**Confidence:** 4

**Summary:**

The paper presents a bottom-up method for 2-D human pose estimation specifically targeting crowded scenes. The main idea is to predict heatmaps around a few anchor candidates (e.g., head, torso, legs), then regress the full body pose from them, followed by pose fusion to get rid of duplicates. It shows good evaluation results on crowded scene datasets.

**Limitations And Societal Impact:**

The authors have adequately addressed the limitations and potential negative societal impact of their work.

**Main Review:**

The paper carefully combines ideas from existing bottom-up methods and designs a system specifically targeting scenes. One key limitation of some existing single-shot systems that operate by only setting an anchor on the center keypoint is that the center may be occluded or overlapping in crowded scenes. Having multiple anchors allows the proposed method to be more robust against such occlusions.

The paper also proposes a carefully crafted pose fusion mechanism which allows mixing and matching keypoint estimates emanating from different anchors, which improves performance.

The reported results on datasets featuring heavily crowded or occluded persons are very good. Results on the more standard COCO dataset (reported in the supplementary material) are also reasonably good but not state of art.

Overall I find that the paper presents a well-engineered human pose estimation system tuned for crowded scenes, which would be more appropriate for a computer vision conference. I think that the paper's methodological advancements and overall novelty is not sufficient enough for publication in NeurIPS.

**Time Spent Reviewing:**

2

---

> ### Author Response · Authors · 2021-08-10
> **Author Response to Reviewer wMdR**
>
> Thanks for your comments! We agree that our paper contains less advancements on machine learning, but it is still within the scope of NeurIPS, which involves both machine learning and computer vision. NeurIPS in previous years also have many computer vision papers that are closely related with ours, i.e., [c] on pose estimation and [a, b, d] on object detection.
>
> Our paper focuses on crowded scenes pose estimation and proposes a novel direct pose-level inference (DPLI) pipeline, which differs with previous top-down and bottom-up pipelines. DPLI relies on visible person parts to distinguish different persons and infers the complete person pose. Compared with previous works, it is free of bounding box detection and keypoint grouping, thus effectively avoids issues of missed person detection and missed keypoint localization. Experimental results show that DPLI is robust to crowded scenes and achieves superior performance. We think our work is important to the pose estimation community and other related computer vision tasks, such as action recognition, where occlusion commonly occurs.
> - - -
> [a] Yihong Chen, Zheng Zhang, Yue Cao, Liwei Wang, Stephen Lin, and Han Hu. Reppoints v2: Verification meets regression for object detection. In NeurIPS, 2020.
>
> [b] Cheng Chi, Fangyun Wei, and Han Hu. Relationnet++: Bridging visual representations for object detection via transformer decoder. In NeurIPS, 2020.
>
> [c] A. Newell, Z. Huang, and J. Deng. Associative embedding: End-to-end learning for joint detection and grouping. In NeurIPS, 2017.
>
> [d] Shaoqing Ren, Kaiming He, Ross Girshick, and Jian Sun. Faster R-CNN: Towards real-time object detection with region proposal networks. In NeurIPS, 2015.

---

### Official Review · Reviewer_AzWE · 2021-07-20

**Rating:** 6
**Confidence:** 4

**Summary:**

This paper presents a deep network method to address occlusions for multi-person pose estimation in a single image. The key idea is to partition each person into several part groups. While the whole body may be occluded, at least one of the part groups may be visible. The proposed network predicts the center of each part group and the offsets towards all the joints for the whole body. The predicted joints from multiple part centers are further refined with dynamic graph convolution network and then fused to make the final prediction. Results are reported on two pose datasets targeted at occluded scenarios with ablation study and comparisons to related works.

**Limitations And Societal Impact:**

It'd be good to present failure cases.

**Main Review:**

Pros:
1) This paper looks at a critical problem of occlusion for multi-person pose estimation in crowed scenes with inter-person or person-object interactions. The proposed part-based multiple pose hypothesis generation is an interesting idea that extends the previous work [15] by X. Nie et al. in ICCV 2019. The idea was first verified in occlusion statistical study (Figure 3) and then the effect of the number of part groups is ablated in Table 1[b].
2) Overall, this paper is well-written. Most of the tech components are explained clearly with a proper amount of math.
3) Results on MSCOCO in supp helps verify the generalization ability of the proposed method.

Cons:
1) The presentation of dynamic graph convolution network is lack of clarity and sufficient details, like how many iterations of updates and how the graph is constructed. The equation (9) is not self-explained. It'd be better to explain its meanings and effects in the context of pose estimation.
2) The proposed network was trained on OCHuman validation set with 2500 images. In Table 2, this paper compared the proposed method with related works on OCHuman test set, but it is hard to tell which compared methods were trained on its validation set. Training on OCHuman validation set seems to be critical for its accuracy. In Table 1(b), the Center-based prediction is pretty similar to [15] without dedicated efforts for handling occlusions but still achieves good results. It'd be good to clarify the training information for Table 2. If the compared methods were not trained or fine-tuned on OCHuman validation, it'd be also interesting to do so and then compare.
3) Table 4 missed an entry from OPEC-Net [21] which achieves 70.6 mAP higher than the proposed method.


**Time Spent Reviewing:**

3 hours

---

> ### Author Response · Authors · 2021-08-10
> **Author Response to Reviewer AzWE**
>
> Thanks for your constructive comments. We clarify the questions as follows.
> - - -
> Q1: More details about dynamic graph convolution network (DGCN) in Pose Refinement.
>
> A1: DGCN is trained and updated with the whole model (backbone and head modules in PPG), no extra updates are required. The graph is an adjacency matrix that reflects the connection of keypoints. For instance, in body skeleton the right wrist is connect to right elbow, thus the value of corresponding element in the matrix is set to 1. After obtaining the initial graph, we set it as parameters of DGCN and updated it with the whole model.
>
> The goal of PR is to extract better person global feature for more accurate pose estimation. Eq. (9) can be viewed as an aggregation process to refine the keypoint features with keypoint relationship. For instance, we can aggregate the wrist feature and shoulder feature to the elbow feature. In this way, we can incorporate pose structure prior to extract a better person global feature.
> - - -
> Q2: Question about the training set in Table 2.
>
> A2: We apologize for the confusion. In fact, all methods in Table 2 are trained on OCHuman validation set containing 2500 images. With the same setting, our method achieves the best performance. We will update corresponding contents to avoid ambiguity.
> - - -
> Q3: Missing methods in Table 4.
>
> A3: We have confirmed with the authors of [21] that the performance of OPEC-Net on CrowdPose is obtained by firstly training on COCO then fine-tuning on CrowdPose. With this setting, it obtains 70.6% mAP. All methods in Table 4 are directly trained on CrowdPose, and they do not use COCO for training. Therefore, OPEC-Net is not cited in Table 4 for fair comparison.
> - - -
> Q4: Presenting failure cases.
>
> A4: Thanks for your suggestion. We will add some failure cases to the manuscript.

---

### Decision · Program_Chairs · 2021-09-27

**Decision:**

Accept (Poster)

**Comment:**

Initially, two of the reviewers expressed concerns about the paper (lack of clarity and limited novelty) and ranked the paper marginally below acceptance. As the ensuing rebuttal managed to successfully address most of reviewer’s concerns, ACs and the majority of the reviewers agreed that this is a strong paper that deserves acceptance. Authors are highly encouraged to address the key comments reported by reviewers as well as to implement all the improvements (as indicated by authors in the rebuttal) in the final camera-ready version.